# NOISES ARE TRANSFERABLE - AN EMPIRICAL STUDY ON HETEROGENEOUS DOMAIN ADAPTATION

## ABSTRACT

Semi-supervised Heterogeneous Domain Adaptation (SHDA) handles the learning of cross-domain samples with both distinct feature representations and distributions. In this paper, we perform the first empirical study on the SHDA problem by utilizing seven typical SHDA approaches for nearly 100 standard SHDA tasks. Surprisingly, we find that the noises drawn from simple distributions as source samples are transferable and can be used to improve the performance of target domain. To go deeper with the essence of the SHDA, we identify and explore several key factors, including the number of source samples, the dimensions of source samples, the original discriminability of source samples, and the transferable discriminability of source samples. Building upon extensive experimental results, we believe that the transferable knowledge in SHDA is primarily rooted in the transferable discriminability of source samples.

## 1 INTRODUCTION

Domain adaptation (DA) (Yang et al., 2020) aims to facilitate the learning task in target domains with only a few or even no labeled samples, by drawing upon knowledge from source domains with sufficient labeled samples. DA techniques have achieved progress in various practical applications (Ge et al., 2020; Peng et al., 2020; Liu et al., 2023; Hoyer et al., 2023). However, most existing DA approaches (Xu et al., 2022; Chen et al., 2019; Zhang et al., 2019; Rangwani et al., 2022) assume that the feature representations of source samples are shared with those of the target ones. Accordingly, they cannot directly handle the *heterogeneous* scenarios, where the source and target samples are characterized by distinct feature representations. These heterogeneous scenarios are both common and significant (Day & Khoshgoftaar, 2017). For instance, the source and target samples come from distinct modalities (Yao et al., 2019; Fang et al., 2023), such as text and image.

To tackle these scenarios, researchers have abstracted and formulated an important but challenging problem, *i.e.*, semi-supervised heterogeneous domain adaptation (SHDA) (Day & Khoshgoftaar, 2017). In the SHDA problem, the source and target samples originate from different feature spaces. Also, the target domain has limited labeled samples and a substantial amount of unlabeled samples available. In addition, there is no one-to-one correspondence *i.e.*, pair information, between cross-domain samples. Numerous SHDA approaches have been developed (Li et al., 2020; Wang et al., 2020; Gu et al., 2022), resulting in improved transfer performance across heterogeneous domains. Although SHDA has achieved great progress, the essential issue, *i.e.*, *what knowledge from a heterogeneous source domain is transferred to target domain?*, has not been well-explored.

To explore the above problem in depth, we conduct extensive experiments utilizing seven typical SHDA approaches (Li et al., 2014; Tsai et al., 2016b; Yao et al., 2020; Chen et al., 2016; Yao et al., 2019; Li et al., 2020; Fang et al., 2023) for nearly 100 standard SHDA tasks. Firstly, we investigate the impact of label and feature information of source samples to target performance. To our surprise, this seemingly significant information is not the dominant factor to influence target performance. Then, according to the above findings, we hypothesize that noises drawn from simple distributions, *e.g.*, Gaussian, Uniform, and Laplace distributions, as source samples may be transferable. Hence, we perform sufficient experiments using noises sampled from Gaussian mixture distributions. As we hypothesized, those noises are indeed helpful and transferable. Finally, to uncover the mysterious veil of transferable knowledge, we proceed with a series of quantitative experiments with different noises to explore several key factors, including the number of source samples, the number of dimensions of

source samples, the original discriminability of source samples, and the transferable discriminability of source samples. Based on experimental results, we hold the perspective that the primary transferred source knowledge is the transferable discriminability of source samples.

We summarize the contributions of this paper as follows. **(1)** To the best of our knowledge, we are the first to execute an empirical study for investigating the nature of the SHDA. **(2)** We identify that the noises drawn from simple distributions as source samples are transferable for the SHDA. **(3)** According to sufficient experimental results, we observe that the transferable discriminability of source samples plays a dominant role in the transferable knowledge of the SHDA.

## 2 RELATED WORK

Existing SHDA methods can be roughly categorized into two sub-branches, *i.e.*, shallow transformation, and deep transformation. To handle the SHDA problem, the former utilizes shallow learning techniques, while the latter relies to deep learning ones.

**Shallow transformation**. Most approaches fall into this sub-branch, primarily utilizing the classifier adaptation and distribution alignment mechanisms for adaptation. Specifically, HFA (Duan et al., 2012), SHFA (Li et al., 2014), and MMDT (Hoffman et al., 2013; 2014) utilize the classifier adaptation mechanism to align the discriminative structures of both domains during adaptation. For example, MMDT transforms the target samples into the source domain by learning a domain-shared support vector machine. A further example is HFA and SHFA, which first augment the transformed samples with the original features and then train a support vector machine shared between domains. LS-UP (Tsai et al., 2016a), PA (Li et al., 2018), SGW (Yan et al., 2018), and KPG (Gu et al., 2022) adopt distribution alignment mechanism to learn optimal transformations. For instance, PA first learns a common space by dictionary-sharing coding, and then alleviates the distributional divergence between domains. Recently, KPG regards labeled cross-domain samples as key samples to guide the correct matching in optimal transport. SCP-ECOC (Xiao & Guo, 2015b), SDASL (Yao et al., 2015), G-JDA (Hsieh et al., 2016), CDLS (Tsai et al., 2016b), SSKMDA (Xiao & Guo, 2015a), DDACL (Yao et al., 2020), and KHDA (Fang et al., 2023) take into account both the classifier adaptation and distribution alignment. As an example, G-JDA and CDLS perform the distribution alignment and classifier adaptation in an iterative manner. Another instance is DDACL, which learns a common classifier by both reducing the distribution discrepancy and enlarging the predition discriminability.

**Deep transformation**. With the advancement of deep learning techniques, some studies have turned to utilizing them to tackle the SHDA problem. Specifically, DTN (Shu et al., 2015) reduces the divergence of the parameters in the last layers across the source and target transformation networks. TNT (Chen et al., 2016) simultaneously considers cross-domain feature transformation, categorization, and adaptation in an end-to-end fashion. Deep-MCA (Li et al., 2019) utilizes a deep neural network to complete the heterogeneous feature matrix and find a better measure function for distribution matching. STN (Yao et al., 2019) adopts the soft-labels of unlabeled target samples to align the conditional distributions across domains, and builds two-layer transformation networks for source and target samples, respectively. SSAN (Li et al., 2020) considers both implicit semantic correlation and explicit semantic alignment in a heterogeneous transfer network. PMGN (Wang et al., 2020) constructs an end-to-end graph prototypical network to learns the domain-invariant class prototype representations, which not only mitigate the distributional divergence but also enhance the prediction discriminability. Recently, JMEA (Fang et al., 2023) jointly trains a transfer classifier and a semi-supervised classifier to acquire high-confidence pseudo-labels for unlabeled target samples.

## 3 PROBLEM FORMULATION

In this section, we introduce the definition of SHDA and give some important terminologies.

The source domain is denoted by $\mathcal{D}_s = \{(\mathbf{x}_i^s, \mathbf{y}_i^s)\}_{i=1}^{n_s}$, where $\mathbf{x}_i^s \in \mathbb{R}^{d_s}$ is the $i$-th source sample represented by $d_s$-dimensional features, and $\mathbf{y}_i^s$ is its corresponding one-hot label over $C$ categories. Similarly, we denote the target domain as $\mathcal{D}_t = \mathcal{D}_l \cup \mathcal{D}_u = \{(\mathbf{x}_i^l, \mathbf{y}_i^l)\}_{i=1}^{n_l} \cup \{\mathbf{x}_i^u\}_{i=1}^{n_u}$, where $\mathbf{x}_i^l$ ($\mathbf{x}_i^u$) $\in \mathbb{R}^{d_t}$ is the $i$-th labeled (unlabeled) target sample with $d_t$-dimensional features, and $\mathbf{y}_i^l$ is its associated one-hot label among $C$ categories. Based on those, the SHDA task is defined as follows.

**Definition 1** *(SHDA). A source domain $\mathcal{D}_s = \{(\mathbf{x}_i^s, \mathbf{y}_i^s)\}_{i=1}^{n_s}$ and a target domain $\mathcal{D}_t = \mathcal{D}_l \cup \mathcal{D}_u = \{(\mathbf{x}_i^l, \mathbf{y}_i^l)\}_{i=1}^{n_l} \cup \{\mathbf{x}_i^u\}_{i=1}^{n_u}$, where the samples in $\mathcal{D}_s$ and $\mathcal{D}_t$ are drawn from distinct distributions, are given, and there is no one-to-one correspondence across them. Also, $d_s \neq d_t$, $n_s \gg n_l$, and $n_u \gg n_l$. The objective of SHDA is to train a model using both $\mathcal{D}_s$ and $\mathcal{D}_t$ to classify samples in $\mathcal{D}_u$.*

## 4 EXPERIMENTAL SETUP

**Datasets**. Following Yao et al. (2019); Li et al. (2020); Wang et al. (2020), we adopt three real-world datasets: **Office+Caltech-10** (Saenko et al., 2010; Griffin et al., 2007), **NUS-WIDE+ImageNet-8** (Chua et al., 2009; Deng et al., 2009), and **Multilingual Reuters Collection** (Amini et al., 2009). The first dataset comprises four domains: Amazon (**A**), Webcam (**W**), and DSLR (**D**), and Caltech-256 (**C**), totaling 10 categories. The second dataset contains two domains of **Text** and **Image** with a total of eight categories. The last dataset includes five domains, *i.e.*, English (**E**), French (**F**), German (**G**), Italian (**I**), and Spanish (**S**), amounting to six categories.

**Pre-process**. We follow (Yao et al., 2019; Li et al., 2020; Wang et al., 2020) to pre-process samples. For the first dataset, we represent images using 800-dimension $SURF$ ($S_{800}$) (Bay et al., 2006) and 4096-dimension $DeCAF_6$ ($D_{4096}$) (Donahue et al., 2014) features. Moreover, we treat all images from source domains as labeled samples. Also, we randomly pick up three images per category in target domains as labeled samples, and the remaining images are regarded as unlabeled samples. For the second dataset, we adopt 64-dimensional features to represent texts, while employ $D_{4096}$ features to characterize images. In addition, we randomly sample 100 texts and three images from each category as labeled samples, while the remaining images are treated as unlabeled samples. For the last dataset, we utilize 1,131, 1,230, 1,417, 1,041, and 807-dimensional reduced features to represent samples from the domains of **E**, **F**, **G**, **I**, and **S**, respectively. Furthermore, we randomly choose 100 articles per category in source domains as labeled samples. Also, we randomly select five and 500 articles from each category as labeled and unlabeled samples, respectively.

**Baselines**. To gain a deeper understanding for the essence of SHDA, we conduct an empirical study using the following baselines: SVMt, NNt, SHFA (Li et al., 2014), CDLS (Tsai et al., 2016b), DDACL (Yao et al., 2020), TNT (Chen et al., 2016), STN (Yao et al., 2019), SSAN (Li et al., 2020), and JMEA (Fang et al., 2023). Among those approaches, the first two approaches are supervised learning methods, while the next seven approaches are SHDA methods. Specifically, SVMt and NNt only utilize the labeled target samples to train a support vector machine and a neural network, respectively. SHFA, G-JDA, and CDLS are the shallow transformation SHDA methods, while TNT, STN, SSAN, JMEA are the deep transformation HDA ones. To ensure a fair comparison, for all baselines, we fix their parameter settings for different SHDA tasks on the same dataset, which are provided in Appendix A.1.

**Metric**. Following Yao et al. (2019); Li et al. (2020); Wang et al. (2020), we adopt the classification accuracy as the evaluation metric. Also, for a fair comparison, we report the average classification accuracy for each approach based on ten random experiments.

## 5 STUDY ON LABEL AND FEATURE INFORMATION OF SOURCE SAMPLES

**Study on label information of source samples**. We investigate how the label information from source samples affects the performance of SHFA. Concretely, we design eight groups of transfer directions: **A** ($S_{800}$) → **C** ($D_{4096}$); **C** ($S_{800}$) → **W** ($D_{4096}$); **W** ($S_{800}$) → **D** ($D_{4096}$); **Text** → **Image**; **E** → **S**; **F** → **S**; **G** → **S**; and **I** → **S**. Here, the first three groups are from the Office+Caltech-10 dataset, with a total of 10 categories. Hence, we construct 10 SHDA tasks for each group by changing the order of category indices for source samples. Specifically, for source samples within the same category, we randomly change their category index to correspond to a distinct category. For instance, if the source samples belonging to category 1, we randomly change their category index to 5. As depicted in Figure 1, the order 1 is the ground-truth order, and the samples from the $i$-th category are sequentially reassigned to other categories, resulting in the shifts in label information. Note that the orders of category indices for target samples follow the ground-truth order, and remain fixed in all tasks. Accordingly, only the SHDA task where the category indices of the source samples follow the order 1 is a vanilla SHDA task. Similarly, we use the same operations to create eight and six SHDA tasks for the fourth and the last four groups, respectively. Hence, we build 62 SHDA tasks in total.

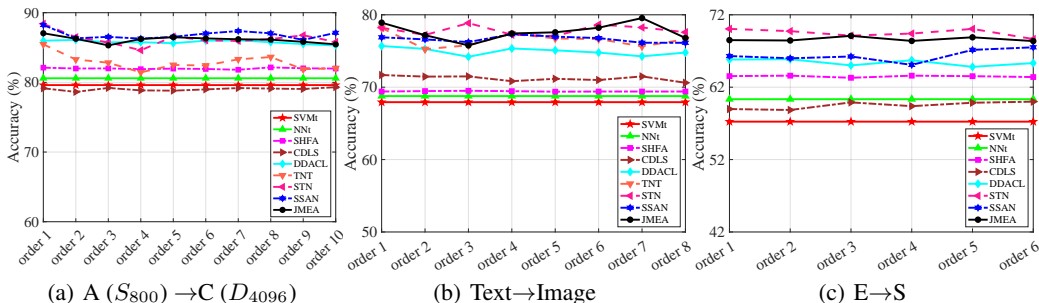

Figure 2: Classification accuracies (%) with distinct orders of category indices for source samples.

Figures 2(a)-2(c) plot the accuracies of all baselines *w.r.t.* distinct orders of category indices for source samples on the tasks of $\mathbf{A}$ $(S_{800}) \rightarrow \mathbf{C}$ $(D_{4096})$, $\mathbf{Text} \rightarrow \mathbf{Image}$, and $\mathbf{E} \rightarrow \mathbf{S}$, respectively. Due to page limit, the experimental results for other tasks are provided in the Appendix A.2.1, and we have similar observations to Figure 2. According to the results, we can observe that as the orders of category indices for source samples change, the accuracies of all methods remain almost unchanged. All the results indicate that the label information of source samples is not the primary factor influencing the performance of SHDA.

| Orders of category indices for source samples | | |
|---|---|---|
| Order 1:  1 2 3 4 5 6 7 8 9 10 | Order 1:  1 2 3 4 5 6 7 8 | Order 1:  1 2 3 4 5 6 |
| Order 2:  2 3 4 5 6 7 8 9 10 1 | Order 2:  2 3 4 5 6 7 8 1 | Order 2:  2 3 4 5 6 1 |
| Order 3:  3 4 5 6 7 8 9 10 1 2 | Order 3:  3 4 5 6 7 8 1 2 | Order 3:  3 4 5 6 1 2 |
| Order 4:  4 5 6 7 8 9 10 1 2 3 | Order 4:  4 5 6 7 8 1 2 3 | Order 4:  4 5 6 1 2 3 |
| Order 5:  5 6 7 8 9 10 1 2 3 4 | Order 5:  5 6 7 8 1 2 3 4 | Order 5:  5 6 1 2 3 4 |
| Order 6:  6 7 8 9 10 1 2 3 4 5 | Order 6:  6 7 8 1 2 3 4 5 | Order 6:  6 1 2 3 4 5 |
| Order 7:  7 8 9 10 1 2 3 4 5 6 | Order 7:  7 8 1 2 3 4 5 6 | |
| Order 8:  8 9 10 1 2 3 4 5 6 7 | Order 8:  8 1 2 3 4 5 6 7 | |
| Order 9:  9 10 1 2 3 4 5 6 7 8 | | |
| Order 10: 10 1 2 3 4 5 6 7 8 9 | | |
| Orders of category indices for target samples | | |
| Order 1:  1 2 3 4 5 6 7 8 9 10 | Order 1:  1 2 3 4 5 6 7 8 | Order 1:  1 2 3 4 5 6 |
| Office+Caltech-10 | NUS-WIDE+ImageNet-8 | Multilingual Reuters Collection |

Figure 1: The orders of category indices for source and target samples on all datasets.

**Study on feature information of source samples**. We investigate how the feature information from source samples influences the performance of SHDA. Since the label information of source samples is not primarily correlated with SHDA performance, we design a series of cross-dataset SHDA tasks. Specifically, we treat the domains of **Image** and **S** as two target domains, each comprising eight and six categories, respectively. For the former, we choose each source domain from the set of {**Text**, $\mathbf{A}$ $(S_{800})$, $\mathbf{C}$ $(S_{800})$, $\mathbf{W}$ $(S_{800})$, $\mathbf{A}$ $(D_{4096})$, $\mathbf{C}$ $(D_{4096})$, $\mathbf{W}$ $(D_{4096})$}. Note that there are a total 10 categories in the domains of $\mathbf{A}$, $\mathbf{C}$, and $\mathbf{W}$, we only utilize the samples belonging to the first eight categories as the source samples. As for the latter, we adopt each domain from the set of {$\mathbf{E}$, $\mathbf{F}$, $\mathbf{G}$, $\mathbf{I}$, $\mathbf{A}$ $(S_{800})$, $\mathbf{C}$ $(S_{800})$, $\mathbf{W}$ $(S_{800})$, $\mathbf{A}$ $(D_{4096})$, $\mathbf{C}$ $(D_{4096})$, **Text**} as the source domain. Analogously, for the domains of $\mathbf{A}$, $\mathbf{C}$, $\mathbf{W}$, and **Text**, we solely employ the samples associated with the first six categories as the source samples. As a result, we create a total of 18 SHDA tasks. Among these tasks, $\mathbf{Text} \rightarrow \mathbf{Image}$, $\mathbf{E} \rightarrow \mathbf{S}$, $\mathbf{F} \rightarrow \mathbf{S}$, $\mathbf{G} \rightarrow \mathbf{S}$, $\mathbf{I} \rightarrow \mathbf{S}$ are vanilla SHDA tasks.

Figures 3(a)-3(b) report the accuracies of all approaches *w.r.t.* different source samples with distinct feature information. Based on the results, we can observe that the accuracy curves of most methods are relatively stable. All the results imply that the feature information of source samples is not the dominant factor affecting the performance of SHDA.

**Study on noises drawn from simple distributions as source samples**. Building upon the above two findings, we hypothesize that utilizing noises sampled from simple distributions as source samples may also yield comparable performance to vanilla SHDA tasks. To confirm this, we generate three noise domains, each with ten, eight, and six categories, based on three Gaussian mixture models. In particular, we directly sample noise from $C$ (*i.e.*, $C = 10, 8, 6$) distinct Gaussian distributions, each with a unique mean vector but the same covariance matrix, representing samples belonging to different categories. Here, $C$ mean vectors are drawn from a standard normal distribution, and the covariance matrix is a $C \times C$ identity matrix. According to the number of categories in those noise domains, we denote them as $\mathbf{N}_{10}$, $\mathbf{N}_8$, and $\mathbf{N}_6$, respectively. Also, for all noise domains, the number of samples in each category is set to 500, with each sample having a dimensionality of 300.

Table 1: Classification accuracy (%) comparison between using noises as source samples and using true source samples.

| $\mathcal{D}_s \to \mathcal{D}_t$ | SVMt | NNt | SHFA | CDLS | DDACL | TNT | STN | SSAN | JMEA |
|---|---|---|---|---|---|---|---|---|---|
| A ($S_{800}$) $\to$ C ($D_{4096}$) | 79.61 | 80.59 | 82.11 | 79.15 | 85.98 | 85.46 | 88.41 | 88.23 | 87.05 |
| $N_{10} \to$ C ($D_{4096}$) | 79.61 | 80.59 | 82.33 | 80.50 | 86.66 | 86.28 | 86.73 | 86.68 | 86.33 |
| C ($S_{800}$) $\to$ W ($D_{4096}$) | 90.11 | 91.66 | 92.19 | 92.19 | 93.21 | 94.87 | 96.87 | 95.70 | 96.30 |
| $N_{10} \to$ W ($D_{4096}$) | 90.11 | 91.66 | 92.15 | 92.91 | 93.92 | 93.43 | 96.38 | 95.32 | 96.91 |
| W ($S_{800}$) $\to$ D ($D_{4096}$) | 93.07 | 93.23 | 93.39 | 94.72 | 94.33 | 92.68 | 95.35 | 95.83 | 95.28 |
| $N_{10} \to$ D ($D_{4096}$) | 93.07 | 93.23 | 93.39 | 93.86 | 94.17 | 94.41 | 94.02 | 94.41 | 93.46 |
| Text $\to$ Image | 67.93 | 68.77 | 69.40 | 71.68 | 75.69 | 78.18 | 78.21 | 76.89 | 78.91 |
| $N_8 \to$ Image | 67.93 | 68.77 | 69.55 | 72.13 | 75.80 | 78.30 | 77.90 | 77.48 | 77.44 |
| E $\to$ S | 57.24 | 60.34 | 63.53 | 58.98 | 65.82 | - | 70.05 | 66.30 | 68.50 |
| $N_6 \to$ S | 57.24 | 60.34 | 63.65 | 63.56 | 63.92 | - | 69.91 | 67.11 | 69.33 |

Accordingly, we construct five SHDA tasks, *i.e.*, $N_{10} \to$ **C** ($D_{4096}$), $N_{10} \to$ **W** ($D_{4096}$), $N_{10} \to$ **D** ($D_{4096}$), $N_8 \to$ **Image**, and $N_6 \to$ **S**.

Table 1 provides a comparison of accuracies of all methods between using noises as source samples and using true source samples. From the results, we can clearly see that all methods using noises as source samples can achieve performance comparable to using true source samples. These observations strongly corroborate that the above hypothesis, *i.e.*, noises sampled from simple distributions as source samples are transferable for SHDA tasks. In other words, those noises contain transferable knowledge.

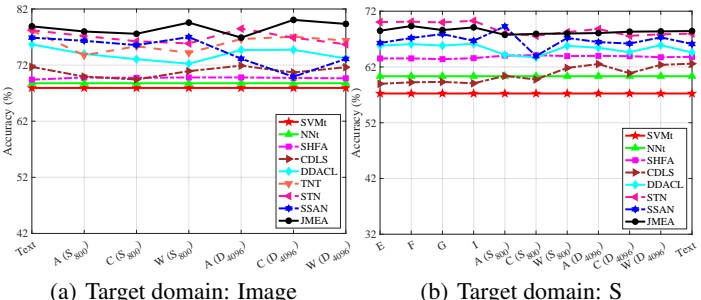

(a) Target domain: Image    (b) Target domain: S

Figure 3: Classification accuracies (%) of different source samples with distinct feature information.

# 6 STUDY ON QUANTITATIVE ANALYSIS OF SOURCE NOISES

Inspired by the above discoveries, we delve further into the investigation of the essence of SHDA. As aforementioned, similar to true source samples, noises as source samples (we refer to them as *source noises* for brevity) also contain transferable knowledge. Accordingly, to attempt to uncover the mysterious veil of transferable knowledge in SHDA, we conduct a series of quantitative experiments with different source noises. Specifically, we first identify four potential sources of transferable knowledge: (i) the number of source samples; (ii) the dimensions of source samples; (iii) the original discriminability of source samples; and (iv) the transferable discriminability of source samples. Then, we consider the domain of **S** as the target domain, and quantitatively analyze the above sources of transferable knowledge by generating different source noises. The experimental details will be presented next.

**Analysis on the number of source samples**. To evaluate how the number of source samples affects the performance of SHDA, we build five SHDA tasks with different numbers of source samples. Concretely, we adopt the noise generation technique described in the previous section to generate five noise domains based on five Gaussian mixture models. For different noise domains, we change the number of samples per category from 300 to 700 with an increment of 100. Moreover, the dimensions of samples in different noise domains are fixed to 300. Based on the number of samples per category in those noise domains, we denote them as $NS_{300}$, $NS_{400}$, $NS_{500}$, $NS_{600}$, and $NS_{700}$, respectively.

Hence, we build a total of five SHDA tasks, *i.e.*, $\mathbf{NS}_{300} \to \mathbf{S}$, $\mathbf{NS}_{400} \to \mathbf{S}$, $\mathbf{NS}_{500} \to \mathbf{S}$, $\mathbf{NS}_{600} \to \mathbf{S}$, and $\mathbf{NS}_{700} \to \mathbf{S}$.

We plot the accuracies of all methods *w.r.t.* the number of source samples in Figure 4(a). From those results, we find that the performance of all methods is almost constant as the number of samples change. These results suggest that the number of source samples is not the primary source of transferable knowledge in SHDA tasks.

**Analysis on the dimensions of source samples**. To assess how the dimension of source samples affects the performance of SHDA, we construct five SHDA tasks with different dimensions of source samples. Specifically, as stated in the previous section, we utilize the same technique to create five noise domains, each sampled from a unique Gaussian mixture model. For different noise domains, the dimensions of samples ranging from 100 to 500 with a step size of 100. In addition, we fix the number of samples per category to 500 across different noise domains. According to the dimensions of samples in those noise domains, we name them as $\mathbf{ND}_{100}$, $\mathbf{ND}_{200}$, $\mathbf{ND}_{300}$, $\mathbf{ND}_{400}$, and $\mathbf{ND}_{500}$, respectively. Therefore, we construct a total of five SHDA tasks, *i.e.*, $\mathbf{ND}_{100} \to \mathbf{S}$, $\mathbf{ND}_{200} \to \mathbf{S}$, $\mathbf{ND}_{300} \to \mathbf{S}$, $\mathbf{ND}_{400} \to \mathbf{S}$, and $\mathbf{ND}_{500} \to \mathbf{S}$.

The accuracies of all baselines *w.r.t.* the dimension of source samples are presented in Figure 4(b). We find that, similar to those results shown in Figure 4(a), under different dimensions of source samples, the performance of all approaches remains relatively stable. The results indicate that the dimension of source samples is also not a dominant source of transferable knowledge in SHDA tasks.

**Analysis on the original discriminability of source samples**. To quantitatively evaluate the original discriminability of source samples, *i.e.*, the discriminability of source samples in the source space, we utilize the Linear Discriminant Analysis (LDA) (Fisher, 1936) as an evaluation metric. That is, we calculate the ratio between-class variance to within-class variance by mapping the source samples into an optimized $(C-1)$-dimensional space, where $C$ is the number of categories. Accordingly, based on the noise generation technique described in the previous section, we generate six noise domains with distinct LDA values. Also, in all noise domains, the dimensions of samples are set to 300, and the number of samples per category is fixed to 500. More specifically, we use two different strategies, *i.e.*, `Category Replicate` and `Category Shift`, to quantitatively control the discriminability of noise domains. In the former strategy, we first generate four noise domains, *i.e.* $\mathbf{NK}_3$, $\mathbf{NK}_4$, $\mathbf{NK}_5$, and $\mathbf{NK}_6$, by adjusting the number of categories, *i.e.*, $K$, from three to six in increments of one. Then, for noise domains where $K$ is less than six, we replicate $6-K$ copies of all samples from a specific category. Subsequently, we assign the remaining class labels to them individually. In the latter strategy, we first sample noises from a Gaussian distribution to represent samples in category one, denoted as $\mathbf{N}_1$. Then, we shift $\mathbf{N}_1$ to generate samples associated with the rest of categories, *i.e.*, $\mathbf{N}_2 = \mathbf{N}_1 + \mathbf{1} \times \lambda$, $\mathbf{N}_3 = \mathbf{N}_1 + \mathbf{1} \times 2\lambda$, $\mathbf{N}_4 = \mathbf{N}_1 + \mathbf{1} \times 3\lambda$, $\mathbf{N}_5 = \mathbf{N}_1 + \mathbf{1} \times 4\lambda$, and $\mathbf{N}_6 = \mathbf{N}_1 + \mathbf{1} \times 5\lambda$. Here, $\mathbf{N}_c$ denotes the samples linked with the category of $c$, and $\mathbf{1}$ denotes an all-one matrix with an appropriate size. Subsequently, we construct two noise domains, *i.e.*, $\mathbf{NL}_{0.40}$ and $\mathbf{NL}_{0.41}$, each with a respective value of $\lambda$ as 0.40 and 0.41. Finally, we create totally six SHDA tasks, *i.e.*, $\mathbf{NK}_3 \to \mathbf{S}$, $\mathbf{NK}_4 \to \mathbf{S}$, $\mathbf{NK}_5 \to \mathbf{S}$, $\mathbf{NK}_6 \to \mathbf{S}$, $\mathbf{NL}_{0.40} \to \mathbf{S}$, and $\mathbf{NL}_{0.41} \to \mathbf{S}$. Also, the LDA values of $\mathbf{NK}_3$, $\mathbf{NK}_4$, $\mathbf{NK}_5$, $\mathbf{NK}_6$, $\mathbf{NL}_{0.40}$, and $\mathbf{NL}_{0.41}$ are 40.51, 51.55, 55.77, 60.57, 57.64, and 60.56, respectively.

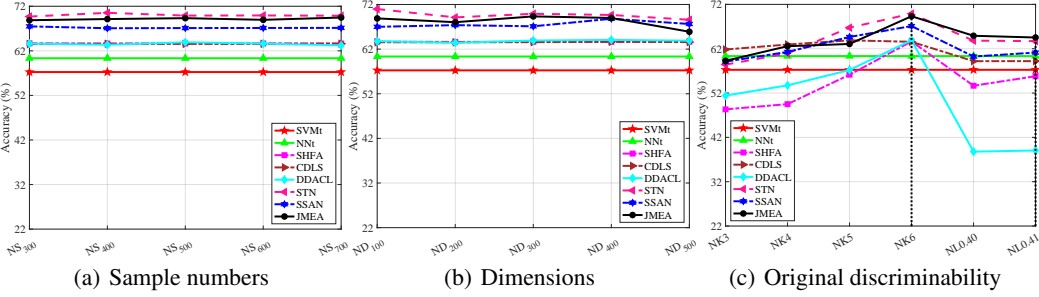

(a) Sample numbers       (b) Dimensions       (c) Original discriminability

Figure 4: Quantitative analysis on the potential sources of transferable knowledge. The target domain is the domain of $\mathbf{S}$.

Figure 4(c) presents the accuracies of all methods *w.r.t.* distinct original discriminability of source samples. We can see that, in the first four tasks, all methods tend to perform better as the LDA values of source domains increases. However, in the last two tasks, although the LDA values of $\mathbf{NL}_{0.40}$ and $\mathbf{NL}_{0.41}$ are close to $\mathbf{NK}_5$ and $\mathbf{NK}_6$, respectively, the performance of most methods on $\mathbf{NL}_{0.40} \rightarrow$ $\mathbf{S}$ and $\mathbf{NL}_{0.41} \rightarrow \mathbf{S}$ is significantly worse than that on $\mathbf{NK}_5 \rightarrow \mathbf{S}$ and $\mathbf{NK}_6 \rightarrow \mathbf{S}$, respectively. This observation suggests that even if the source domain itself has a good discriminability, all methods may still achieve poor performance. One possible reason is that the source samples, generated via the `Category Shift` strategy, may have lost a significant amount of transferable knowledge. Thus, those results imply that the primary source of transferable knowledge in SHDA tasks does not lie in the original discriminability of source samples.

**Analysis on the transferable discriminability of source samples**. We ultimately focus our attention on the `transferable discriminability` of source samples, *i.e.*, the transferability and discriminability of source samples in the common space. To quantitatively characterize those properties, we design a simple Heterogeneous Classification Network (HCN), which projects labeled samples from all domains into a common space by training a transferable classifier. Specifically, the objective function for HCN is formulated as

$$\min_{f,g_s,g_t} \frac{1}{n_l} \sum_{i=1}^{n_l} \mathcal{L}\big(\mathbf{y}_i^l, f(g_t(\mathbf{x}_i^l))\big) + \beta \underbrace{\frac{1}{n_s} \sum_{i=1}^{n_s} \mathcal{L}\big(\mathbf{y}_i^s, f(g_s(\mathbf{x}_i^s))\big)}_{\mathcal{R}_s} + \tau\big(\|g_s\|^2 + \|g_t\|^2 + \|f\|^2\big), \quad (1)$$

where $\mathcal{L}(\cdot, \cdot)$ is the cross-entropy function, $g_t(\cdot)$ and $g_s(\cdot)$ are two single-layer fully connected networks with the Leaky ReLU activation functions (Maas et al., 2013), respectively, $f(\cdot)$ is the softmax classifier, $\mathcal{R}_s$ is the empirical risk of source samples, $\beta$ is a positive trade-off parameter to adjust the importance of $\mathcal{R}_s$, and $\tau$ is a positive regularization parameters for preventing over-fitting. Note that it is only meaningful to measure transferable discriminability of source samples while avoiding over-fitting. Thus, we need to carefully tune the parameters of the HCN. Also, for a fair comparison, we solely tune the parameters of the HCN on one task to find the optimal parameter setting, *i.e.*, $\beta = 0.1$ and $\tau = 0.005$, and then apply it to other tasks.

By optimizing problem (1), we can learn the optimized $g_s(\cdot)$ and $g_t(\cdot)$ that map source and target samples into an optimized common space, respectively. Subsequently, inspired by (Long et al., 2013; Tsai et al., 2016b; Hsieh et al., 2016; Yao et al., 2019; Fang et al., 2023), in the common space, we utilize the projected Maximum Mean Divergence (MMD) with linear kernel to characterize the transferability of source samples $\mathcal{T}_s$, which is calculated by

$$\mathcal{T}_s = \frac{1}{C+1} \sum_{c=0}^{C} \big\| \frac{1}{n_s^c} \sum_{i=1}^{n_s^c} g_s(\mathbf{x}_{i,c}^s) - \frac{1}{n_l^c} \sum_{i=1}^{n_l^c} g_t(\mathbf{x}_{i,c}^l) \big\|^2, \quad (2)$$

where $\mathbf{x}_{i,c}^s$ and $\mathbf{x}_{i,c}^l$ denote the $i$-th source and labeled target sample associated in category $c$, respectively, $n_s^c$ and $n_l^c$ denote the number of source and labeled target samples belonging to category $c$, respectively, and $C$ denotes the number of categories. Note that, for clarity, we assign all source and labeled target samples to the 0-th category. That is, $n_s^0 = n_s$, $n_l^0 = n_l$, $\mathbf{x}_{i,0}^s = \mathbf{x}_i^s$, and $\mathbf{x}_{i,0}^l = \mathbf{x}_i^l$. The smaller $\mathcal{T}_s$ is, the more similar the distributions of source and target domains, and the better the transferability of source samples. In addition, we employ the empirical risk of source samples, *i.e.*, $\mathcal{R}_s$, to characterize the discriminability of source samples in the common space. The smaller it is, the higher the discriminability of source samples. Putting them together, we form an evaluation metric of the transferable discriminability for source samples $\mathcal{TR}_s$, which is defined as

$$\mathcal{TR}_s = \mathcal{T}_s + \mathcal{R}_s, \quad (3)$$

where a smaller value of $\mathcal{TR}_s$ indicates a higher transferable discriminability of source samples.

Since $\mathcal{TR}_s$ is measured in the common space, it is difficult to acquire different values of $\mathcal{TR}_s$ by controlling the generation of source samples in the source space. To examine the impact of $\mathcal{TR}_s$ to the performance of SHDA, we calculate its values for all tasks in the three aforementioned experiments. Table 2 reports the transferable discriminability of source samples *w.r.t.* the number of samples and dimension. Moreover, we also list the average classification accuracy of seven SHDA baselines, *i.e.*, $\text{AVG}_{acc}$, and the classification accuracy of HCN, *i.e.*, $\text{HCN}_{acc}$. We can observe that the $\mathcal{TR}_s$ values of all tasks are relatively close and small, which implies that source

Table 2: Transferable discriminability of source samples with distinct sample numbers and dimensions, and the target domain is the domain of **S**. Here, $\mathcal{TR}_s$ denotes the transferable discriminability of source samples, $\text{AVG}_{acc}$ denotes the average classification accuracy of seven SHDA baselines, and $\text{HCN}_{acc}$ denotes the classification accuracy of HCN.

| | $\text{NS}_{300}$ | $\text{NS}_{400}$ | $\text{NS}_{500}$ | $\text{NS}_{600}$ | $\text{NS}_{700}$ | $\text{ND}_{100}$ | $\text{ND}_{200}$ | $\text{ND}_{300}$ | $\text{ND}_{400}$ | $\text{ND}_{500}$ |
|---|---|---|---|---|---|---|---|---|---|---|
| $\mathcal{TR}_s$ | 0.27 | 0.27 | 0.27 | 0.28 | 0.27 | 0.31 | 0.28 | 0.27 | 0.27 | 0.26 |
| $\text{AVG}_{acc}$ (%) | 66.15 | 66.22 | 66.25 | 66.15 | 66.16 | 66.31 | 65.85 | 66.25 | 66.46 | 65.54 |
| $\text{HCN}_{acc}$ (%) | 56.10 | 55.79 | 56.07 | 55.87 | 56.00 | 55.88 | 56.30 | 56.07 | 56.24 | 55.94 |

domains in those tasks exhibit the similar and superior transferable discriminability. Also, it is a significant reason why the values of $\text{AVG}_{acc}$ of all tasks are close to each other and significantly exceed those of $\text{HCN}_{acc}$. Similar to Table 2, Table 3 lists those values *w.r.t.* different original discriminability of source samples. In addition, we also present the original discriminability of source samples using LDA values for comparison. We have the following insightful observations. (1) For the source domains of $\mathbf{NK}_3$, $\mathbf{NK}_4$ $\mathbf{NK}_5$, and $\mathbf{NK}_6$, with the increase of the original discriminability of source samples, the values of $\mathcal{TR}_s$ gradually decrease, indicating improvement in the transferable discriminatibility of source samples. This explains why the $\text{AVG}_{acc}$ values progressively increase over the four tasks. (2) For the source domains of $\mathbf{NL}_{0.40}$ and $\mathbf{NL}_{0.41}$, while their original discriminability is relatively high, their transferable discriminability is somewhat low, leading to poor performance. This indicates that there is no clear correspondence between the original and transferable discriminability of source samples. (3) The value of $\mathcal{TR}_s$ for the $\mathbf{NK}_6 \rightarrow \mathbf{S}$

Table 3: Transferable discriminability of source samples with distinct original discriminability, and the target domain is the domain of **S**. Here, LDA denotes original discriminability of source samples, $\mathcal{TR}_s$ denotes the transferable discriminability of source samples, $\text{AVG}_{acc}$ denotes the average classification accuracy of seven SHDA baselines, and $\text{HCN}_{acc}$ denotes the classification accuracy of HCN.

| | $\text{NK}_3$ | $\text{NK}_4$ | $\text{NK}_5$ | $\text{NK}_6$ | $\text{NL}_{0.40}$ | $\text{NL}_{0.41}$ |
|---|---|---|---|---|---|---|
| LDA | 40.51 | 51.55 | 55.77 | 60.57 | 57.64 | 60.56 |
| $\mathcal{TR}_s$ | 1.87 | 1.22 | 0.69 | 0.27 | 1.73 | 1.72 |
| $\text{AVG}_{acc}$ (%) | 56.39 | 58.50 | 61.95 | 66.25 | 56.77 | 57.24 |
| $\text{HCN}_{acc}$ (%) | 56.25 | 56.31 | 55.81 | 56.07 | 55.74 | 55.56 |

task is the smallest, and the average accuracy of all baselines is the highest. This suggests that $\mathcal{TR}_s$ can assess the quality of source noises to some extent.

In addition, via t-SNE (Van der Maaten & Hinton, 2008), Figure 5 shows the visualization results on the tasks of $\mathbf{NK}_6 \rightarrow \mathbf{S}$ and $\mathbf{NL}_{0.41} \rightarrow \mathbf{S}$. The results offer several meaningful observations. (1) For the domains of $\mathbf{NK}_6$ and $\mathbf{NL}_{0.41}$, samples from both domains are well separable in their original spaces, which exhibits their good original discriminability. (2) For the task of $\mathbf{NK}_6 \rightarrow \mathbf{S}$, in the common space, the distribution of source samples is consistent with that of labeled target samples, and the source samples shows better discriminability. This shows good transferable discriminability in the domain of $\mathbf{NK}_6$. (3) For the task of $\mathbf{NL}_{0.41} \rightarrow \mathbf{S}$, in the common space, the distributions of source and labeled target samples are observably distinct, and the source samples do not exhibit superior separability. This exhibits poor transferable discriminability in the domain of $\mathbf{NL}_{0.41}$. The visualization results on other tasks are included in Figure 8 in Appendix A.2.2. All the results indicate that there is no evident correlation between the original and transferable discriminability of source samples, and the transferable discriminability of source samples is important for target performance.

Building upon all the above results, we believe that the principal source of transferable knowledge in SHDA tasks is rooted in the transferable discriminability of source samples. In addition, it can be used as a metric to guide the generation of noises in practical applications, which is one possible future direction of our work.

In summary, we highlight the following major findings:

- The primary source of transferable knowledge in SHDA tasks does not lie in the following factors, *i.e.*, the number of source samples, the dimension of source samples, and the original discriminability of source samples.

- The transferable discriminability of source samples is a dominant source of transferable knowledge in SHDA tasks.

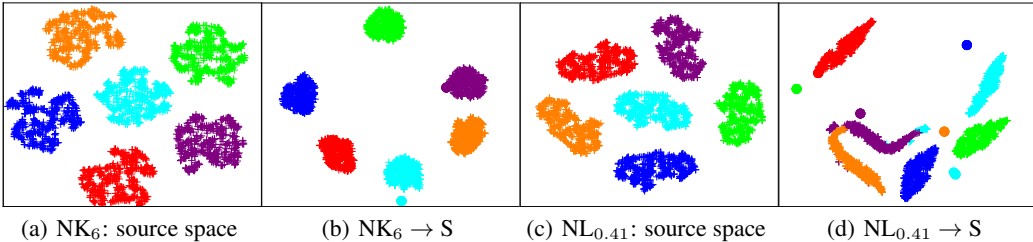

(a) $NK_6$: source space  (b) $NK_6 \rightarrow S$  (c) $NL_{0.41}$: source space  (d) $NL_{0.41} \rightarrow S$

Figure 5: t-SNE visualization. Here, the '+' sign denotes source sample, while the '•' sign represents labeled target sample. Each color corresponds to a distinct category.

## 7 STUDY ON DISTRIBUTIONS OF SOURCE NOISES

The experiments above all employ source noises drawn from Gaussian distributions. To compare the impact of using different types of distributions on the performance of SHDA, we conduct the study on different types of distributions in Appendix A.2.3. The results indicate that using distinct types of distributions has a relatively minor impact on performance, and the Gaussian distribution is better.

## 8 DISCUSSION

In the experiments above, a crucial discovery is that noises are useful in the SHDA problem, which seems a bit counter-intuitive. In reality, however, several studies (Baradad Jurjo et al., 2021; Tang et al., 2022; Luo et al., 2021) have paid attention to the value of noises for tackling machine learning tasks. In the following, we compare this discovery with some related studies, and highlight its value in practical applications.

**Comparison with related studies**. To our knowledge, there are some studies (Baradad Jurjo et al., 2021; Luo et al., 2021; Tang et al., 2022) closely related to ours. Baradad Jurjo et al. (2021) utilize noises to deal with the representation learning problem. They pre-tain deep networks by using noises generated from several simple processes. Their experiments demonstrate that those noises could effectively enhance the learning of visual representations. Luo et al. (2021) adopt noises to handle the non-iid problem in federated learning (FL). They first estimate the global mean and covariance information for each category. Then, based on such information, they sample noises from an approximated Gaussian mixture distribution to fine-turn the classifier. Their experiments reveal that those noises substantially improve the classification performance. Similar to (Luo et al., 2021), Tang et al. (2022) also employ noises to tackle non-iid issue in FL. They first upsample pure Gaussian noises, and then align the distributions of noises and vanilla samples in each client. Their experiments prove that FL could significantly benefit from those noises. In summary, these studies match with our discovery to some extent, but they do not perform empirical analysis on noises.

**Value in practical applications**. Vanilla homogeneous/heterogeneous DA methods (Pan & Yang, 2010; Csurka, 2017; Zhuang et al., 2020; Day & Khoshgoftaar, 2017) assume that the source samples are publicly available. However, in many practical applications, it is often not easy to acquire those samples due to privacy, confidentiality and copyright issues. Our discovery offers a potential solution to those issues. That is, noises sampled from simple distributions could be used as source samples.

## 9 CONCLUSION

In this paper, we conduct extensive experiments to explore the essence of the SHDA. We first observe that noises sampled from the different distributions as source samples are beneficial to the SHDA. After this, we proceed with a series of quantitative analysis experiments by generating different noises. Based on extensive experimental results, we hold an opinion that the transferable discriminability of source samples is the dominant factor affecting the performance of the SHDA. We believe that the above findings provide a new perspective for the SHDA. Moreover, applying noises to facilitate other learning tasks is our future direction.

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

# A APPENDIX

## A.1 PARAMETER SETTINGS

In this paper, we utilize the following baselines: SVMt, NNt, SHFA (Li et al., 2014), CDLS (Tsai et al., 2016b), DDACL (Yao et al., 2020), TNT (Chen et al., 2016), STN (Yao et al., 2019), SSAN (Li et al., 2020), JMEA (Fang et al., 2023). To ensure a fair comparison, for each baseline, we fix its parameter setting across different SHDA tasks with the same dataset. The detailed parameter settings for all approaches are listed as follows.

**SVMt**[1]. We utilize LIBSVM (Chang & Lin, 2011) to run SVMt, which solely utilizes the labeled target samples to learn a support vector machine. The regularization parameter $C$ (see Eq. (1) in (Chang & Lin, 2011)) is set to 1.

**NNt**. We implement NNt based on the TensorFlow framework (Abadi et al., 2016). The objective function of NNt is formulated as

$$\min_{f, g_t} \frac{1}{n_l} \sum_{i=1}^{n_l} \mathcal{L}\big(\mathbf{y}_i^l, f(g_t(\mathbf{x}_i^l))\big) + \tau\big(\|g_t\|^2 + \|f\|^2\big), \tag{4}$$

where $\mathcal{L}(\cdot, \cdot)$ is the cross-entropy function, $g_t(\cdot)$ is a single-layer fully connected networks with the Leaky ReLU activation functions (Maas et al., 2013), and $f(\cdot)$ is the softmax classifier. We optimize Eq. (4) by utilizing the Adam optimizer (Kingma & Ba, 2015) with a learning rate of 0.01, and empirically set $\tau = 0.001$. Also, the dimension of hidden layer representations is set to 256, and the number of iterations is specified as 100.

**SHFA**[2]. It first augments transformed samples with original ones, and then learns a support vector machine in a semi-supervised manner. For all tasks, we employ the default parameter settings described in Section 4.1 of (Li et al., 2014), and the parameter $\lambda$ is empirically fixed to 1 (see Section 4.1 in (Li et al., 2014)).

**CDLS**[3]. It identifies representative cross-domain samples during distribution alignment. The recommended parameter settings detailed in Section 4.1 of (Tsai et al., 2016b) are used on all tasks.

**DDACL**[4]. It learns a softmax classifier by both aligning the distributions across domains and enlarging the discriminability of cross-domains samples. As described in Section 5.1 in (Yao et al., 2020), we utilize the default parameter settings for all tasks, and the parameter $\tau$ is empirically set to 0.001.

**TNT**[5]. It jointly solves feature transformation, distribution alignment, and label prediction in a unified neural network architecture. For all tasks, we follow the suggested parameter settings outlined in Section 4.1 of (Chen et al., 2016).

**STN**[6]. It adopts the soft-labels of unlabeled target samples to reduce the conditional distributional divergence across domains, and jointly learns a transferable classifier and a common space in an end-to-end fashion. Following (Yao et al., 2019), we utilize the default parameter settings on all tasks.

**SSAN**[7]. It learns a heterogeneous transfer network by taking implicit semantic correlation and explicit semantic alignment into account. As presented in Section 4.1 in (Li et al., 2020), we employ the recommended parameter settings for all tasks, and the number of epochs is set to 1000.

**JMEA**[8]. It simultaneously learns a transferable classifier and a semi-supervised classifier to acquire high-confident pseudo-labels for unlabeled target samples. For all tasks, we adopt the suggested parameter settings in Section 8.2 of (Fang et al., 2023) except for the parameter $\rho$ (see Algorithm 2 in (Fang et al., 2023)). For tasks derived from the Office+Caltech-10 and Multilingual Reuters Collection

---

[1]https://www.csie.ntu.edu.tw/ cjlin/libsvm/

[2]https://github.com/wenli-vision/SHFA_release

[3]https://github.com/yaohungt/CrossDomainLandmarksSelectionCDLS/tree/master

[4]https://github.com/yyyaoyuan/DDA

[5]https://github.com/wyharveychen/TransferNeuralTrees

[6]https://github.com/yyyaoyuan/STN

[7]https://github.com/BITDA/SSAN

[8]https://github.com/fangzhen/SemisupervisedHeterogeneousDomainAdaptation

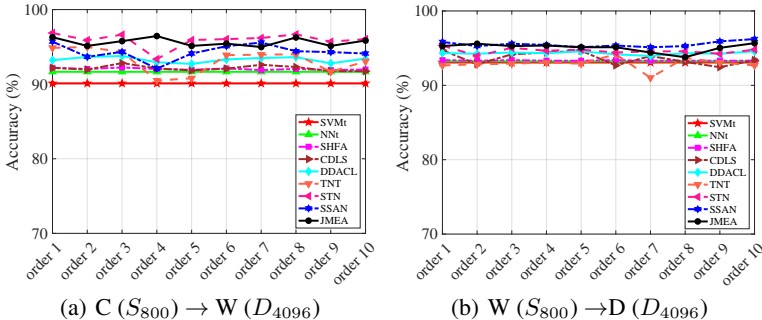

Figure 6: Classification accuracies (%) with distinct orders of category indices for source samples on the Office+Caltech10 dataset.

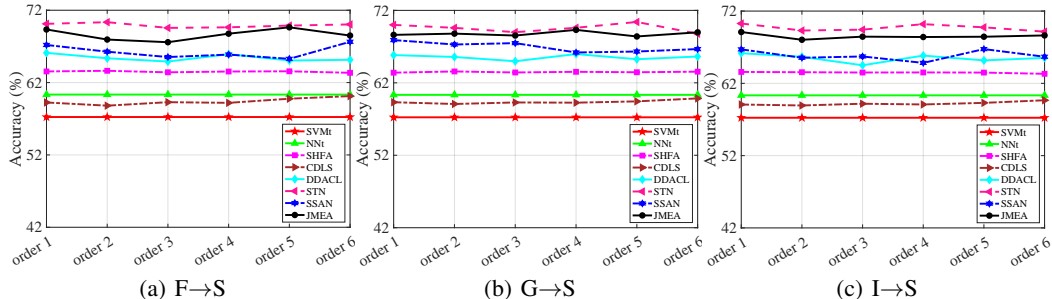

Figure 7: Classification accuracies (%) with distinct orders of category indices for source samples on the Multilingual Reuters Collection dataset.

datasets, we empirically set $\rho$ to be 0.0001. As for tasks derived from the NUS-WIDE+ImageNet-8 dataset, $\rho$ is empirically set to 0.001.

## A.2 MORE EXPERIMENTAL RESULTS

### A.2.1 RESULTS OF LABEL INFORMATION FOR SOURCE SAMPLES ON OTHER TASKS

Figures 6-7 show the accuracies of all baselines *w.r.t.* distinct orders of category indices for source samples on the tasks of $\mathbf{C}\ (S_{800}) \to \mathbf{W}\ (D_{4096})$, $\mathbf{W}\ (S_{800}) \to \mathbf{D}\ (D_{4096})$, $\mathbf{F} \to \mathbf{S}$, $\mathbf{G} \to \mathbf{S}$, and $\mathbf{I} \to \mathbf{S}$, respectively. From those results, we can see that as change in the orders of category indices for source samples, the accuracies of all methods remain largely consistent.

### A.2.2 T-SNE VISUALIZATION RESULTS ON OTHER TASKS

Figure 8 presents the t-SNE (Van der Maaten & Hinton, 2008) visualization results on the tasks of $\mathbf{NK}_3 \to \mathbf{S}$, $\mathbf{NK}_4 \to \mathbf{S}$, $\mathbf{NK}_5 \to \mathbf{S}$, and $\mathbf{NL}_{0.40} \to \mathbf{S}$. We have the following insightful observations. (1) For the domains of $\mathbf{NK}_3$, $\mathbf{NK}_4$, and $\mathbf{NK}_5$, there are mainly three, four, and five clusters in their source spaces, respectively. This is caused by the `Catagory Replicate` strategy. In addition, similar phenomena occur in the common spaces, indicating that overlapping samples are difficult to separate. Also, since labeled target samples have better discriminability, source domains with poorer discriminability result in larger distributional divergence across domains. (2) For the domain of $\mathbf{NL}_{0.40}$, samples exhibit superior discriminability in the source space. However, in the common space, they are not as well separable as in the source space. Also, the distribution of source samples is significantly differs from that of labeled target samples. This verifies that there is no clear correspondence across the original and transferable discriminability of source samples again.

### A.2.3 STUDY ON DISTRIBUTIONS OF SOURCE NOISES

In the above experiments, all source noises drawn from Gaussian distributions. To compare the impact of using different types of distributions on the performance of SHDA, we establish three noise domains, *i.e.*, $\mathbf{NG}$, $\mathbf{NU}$, and $\mathbf{NL}$, based on Gaussian, Uniform, and Laplace distributions, respectively.

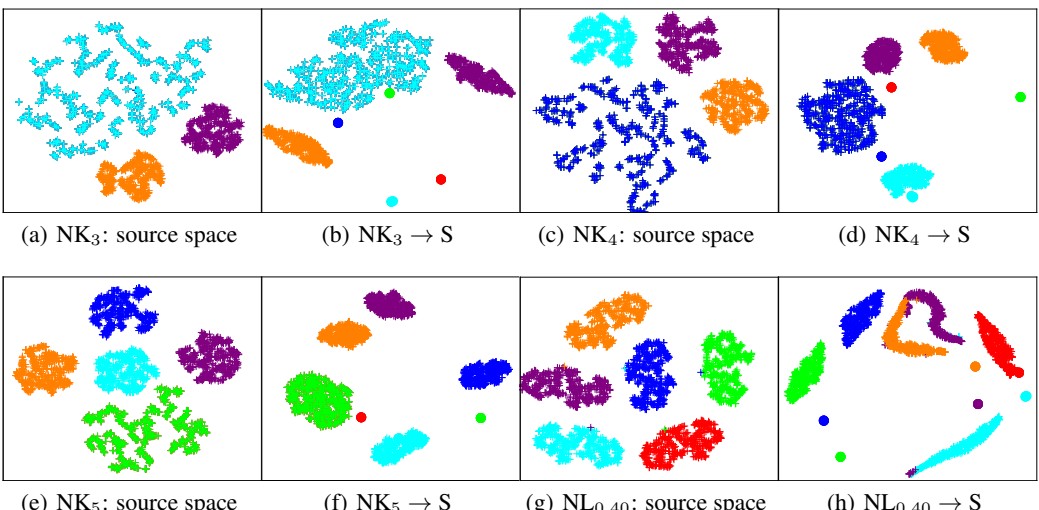

| (a) NK$_3$: source space | (b) NK$_3 \rightarrow$ S | (c) NK$_4$: source space | (d) NK$_4 \rightarrow$ S |

| (e) NK$_5$: source space | (f) NK$_5 \rightarrow$ S | (g) NL$_{0.40}$: source space | (h) NL$_{0.40} \rightarrow$ S |

Figure 8: t-SNE visualization. Here, the '+' sign denotes source sample, while the '•' sign represents labeled target sample. Each color corresponds to a distinct category.

For a fair comparison, in all noise domains, we fix the number of samples within each category to 100, and the dimensions of samples are set to 300. In particular, we employ the noise generation technique detailed in the previous section to create the **NG** domain. For the construction of the **NU** domain, we sample samples per category from $Uniform(-10, 10)$. We build the **NL** domain by sampling samples within each category from $Laplace(0, 1)$. As a result, we create three SHDA tasks, *i.e.*, **NG** $\rightarrow$ **S**, **NU** $\rightarrow$ **S**, and **NL** $\rightarrow$ **S**. According to the results presented in Figure 9, we can observe that using different kinds of distributions has a relatively minor impact on the performance of all methods. In addition, Table 4 lists the transferable discriminability of source samples *w.r.t.* distinct distributions, as well as the average classification accuracy of seven SHDA baselines and the classification accuracy of HCN. We find that utilizing Uniform and Laplace distributions results in slightly worse transferable discriminability than using Gaussian distribution. Building upon this finding, we recommend employing Gaussian distribution in practical applications, as it may yield better performance.

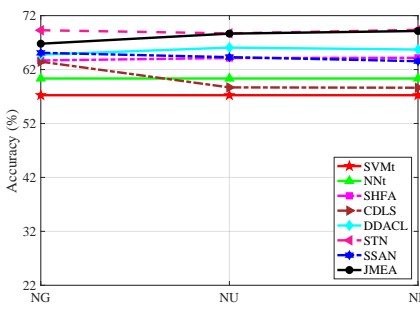

Figure 9: Classification accuracies (%) with different distributions of source samples. The target domain is the domain of S.

Table 4: Transferable discriminability of source samples with distinct distributions, and the target domain is the domain of **S**. Here, $\mathcal{TR}_s$ denotes the transferable discriminability of source samples, AVG$_{acc}$ denotes the average classification accuracy of seven SHDA baselines, and HCN$_{acc}$ denotes the classification accuracy of HCN.

|  | NG | NU | NL |
|---|---|---|---|
| $\mathcal{TR}_s$ | 0.26 | 0.55 | 0.54 |
| AVG$_{acc}$ (%) | 65.52 | 65.10 | 65.10 |
| HCN$_{acc}$ (%) | 55.88 | 55.73 | 55.81 |

