# OpenReview forum: "Noises are Transferable - An Empirical Study on Heterogeneous Domain Adaptation"
_ICLR.cc/2024/Conference — ICLR 2024 Conference Withdrawn Submission_

### Official Review · Reviewer_9AV3 · 2023-10-31

**Soundness:** 2 fair
**Presentation:** 4 excellent
**Contribution:** 2 fair
**Rating:** 5
**Confidence:** 4

**Summary:**

In this paper, the authors investigate the problem of semi-supervised heterogeneous domain adaptation, where the source and target are characterized by different feature representations. To explore which information can be transferred in heterogeneous domain adaptation, the authors conduct expensive experiments on different heterogeneous domain adaptation benchmarks and find that the noise is transferable.

**Strengths:**

The authors conduct expensive experiments on the mainstream heterogeneous domain adaptation datasets and find some interesting conclusions.

**Weaknesses:**

Weak Points:
1.	Although the authors draw some interesting results, where the noise can be transferred from the source to the target domain, I think this result is still counterintuitive. If the noise is transferable in semi-supervised heterogeneous domain adaptation, it also can be transferred in the unsupervised heterogeneous domain adaptation, it is unclear why the authors limit this strong conclusion in the semi-supervised scenarios.
2.	According to Section 5, the authors claim that the label information of the source sample might not be the primary factor that influences the performance of SHDA. Since the authors conduct the experiments with some large models like JMEA. It employs the ResNet-50, a very deep pre-trained network, which might contain some label information. Therefore, it cannot sufficiently reflect that the label information is useless. It is suggested that the authors should employ a lighter neural architecture like AlexNet to evaluate the proposed idea.
3.	Since the authors evaluate the performance in the semi-supervised scenario, it is suggested that the authors should provide the experiment results of training with labeled target data. Because I doubt that some target-labeled data with a reasonable learning rate (to avoid overfitting) might be enough for the ideal performance.
4.	In the part ‘Study on feature information of source samples’, the authors use features with different dimensions to denote different information, for example, D(4096) contains more information than D(800). This might be unreasonable since features with different dimensions might contain equal information.

**Questions:**

N.A.

---

> ### Author Response · Authors · 2023-11-20
> **Response to Reviewer 9AV3**
>
> Thanks very much for your valuable comments and questions. We address your concerns as follows.
>
> > W1. Although the authors draw some interesting results, where the noise can be transferred from the source to the target domain, I think this result is still counterintuitive. If the noise is transferable in semi-supervised heterogeneous domain adaptation, it also can be transferred in the unsupervised heterogeneous domain adaptation, it is unclear why the authors limit this strong conclusion in the semi-supervised scenarios.
>
> **A.** This is a good question. **Although noises are transferable, they cannot be directly applied in unsupervised scenarios**. Indeed, **we generate source noises based solely on the number of categories**. For instance, let's assume that there are 10 categories in the target domain. We sample noises from 10 distinct Gaussian distributions and treat them as samples belonging to different categories. Accordingly, **those noises are task-agnostic, and their categories have no practical meaning**. Hence, **to avoid randomness in the learning process of a specific target task, having a small number of labeled target samples is essential. Also, their category indices are used to establish connections with those of noises**. Based on our findings, we can utilize domain adaptation mechanisms to jointly train all samples, including noises, and labeled and unlabeled target samples, to enhance target performance.
>
> >W2. According to Section 5, the authors claim that the label information of the source sample might not be the primary factor that influences the performance of SHDA. Since the authors conduct the experiments with some large models like JMEA. It employs the ResNet-50, a very deep pre-trained network, which might contain some label information. Therefore, it cannot sufficiently reflect that the label information is useless. It is suggested that the authors should employ a lighter neural architecture like AlexNet to evaluate the proposed idea.
>
> **A.** We feel that there may be a misunderstanding about the baselines. In the SHDA problem, mainstream deep transformation approaches, including TNT, STN, SSAN, and JMEA, typically follow a two-step process. They first pre-process source and target samples by extracting heterogeneous features, e.g., $S_{800}$ and $D_{4096}$ (please refer to the Experimental Setup section in our manuscript). Then, to perform heterogeneous domain adaptation, **they train lightweight neural network architectures from scratch**. Accordingly, **JMEA does not contain any label information**. Also, we have used lightweight neural network architectures for experiments. Thus, based on the experimental results, we can infer that the label information of the source sample is not the primary factor influencing the performance of SHDA.
>
> >W3. Since the authors evaluate the performance in the semi-supervised scenario, it is suggested that the authors should provide the experiment results of training with labeled target data. Because I doubt that some target-labeled data with a reasonable learning rate (to avoid overfitting) might be enough for the ideal performance.
>
> **A.** **We have reported the performance of supervised learning baselines, i.e., SVMt and NNt**. Specifically, SVMt and NNt only utilize the labeled target samples to train a support vector machine and a neural network, respectively (please refer to the Experimental Setup section in our manuscript). Also, in our manuscript, we can clearly see from Table 1 that, most SHDA methods are far better than SVMt and NNt. Specifically, on the task of A ($S_{800}$) $\rightarrow$ C ($D_{4096}$), the classification accuracy of STN is 88.41\%, which substantially exceeds the best supervised method, i.e., NNt, by 7.82\%. Furthermore, **it is difficult to achieve ideal performance solely through supervised learning approaches, which can be confirmed in many SHDA studies**.
>
> >W4. In the part ‘Study on feature information of source samples’, the authors use features with different dimensions to denote different information, for example, D(4096) contains more information than D(800). This might be unreasonable since features with different dimensions might contain equal information.
>
> **A.** In our experiments, **not only do the dimensions differ, but the meaning of each dimension is also distinct**. Specifically, we follow the stand SHDA settings and adopt two different features, i.e., $S_{800}$ and $D_{4096}$, to represent images (please refer to the Experimental Setup section in our manuscript). They are completely different feature representations. Also, **we do not use the $D_{800}$ features to represent images**.

---

### Official Review · Reviewer_WVvA · 2023-11-01

**Soundness:** 3 good
**Presentation:** 3 good
**Contribution:** 3 good
**Rating:** 6
**Confidence:** 2

**Summary:**

This paper conducts a comprehensive empirical study on the SHDA problem. The findings reveal that noise, when sampled from simple distributions as source data, can be transferable. Furthermore, the study identifies the transferable discriminability of source samples as the key factor in the knowledge transfer of SHDA.

**Strengths:**

1.	The empirical study is very extensive.
2.	This paper reports a surprising finding that noise when sampled from simple distributions as source data, can be transferable.

**Weaknesses:**

Could you clarify the distinction between Semi-supervised Heterogeneous Domain Adaptation (SHDA) and Semi-supervised Domain Adaptation (SSDA) according to Definition 1? Both seem to align with Definition 1. It appears that $d_s$ and $d_t$ merely represent specific data dimensions and may not capture the heterogeneity in the nature or type of features between the source and target domains.

**Questions:**

• What was the rationale behind using fixed parameter settings for different SHDA tasks on the same dataset? Doesn't this approach risk not capturing the optimal performance for each task?

• In the section of analysis on the original discriminability of source samples, what led to the choice of $\lambda=0.4$ and $\lambda=0.41$?

• In the section of analysis on the transferable discriminability of source samples, when using $g_t(\cdot)$ as a single layer fully connected networks with the Leaky ReLU, is there any potential for underfitting the data?

---

> ### Author Response · Authors · 2023-11-20
> **Response to Reviewer WVvA**
>
> Thanks very much for your valuable comments and questions. We address your concerns as follows.
>
> >W1. Could you clarify the distinction between Semi-supervised Heterogeneous Domain Adaptation (SHDA) and Semi-supervised Domain Adaptation (SSDA) according to Definition 1? Both seem to align with Definition 1. It appears that $d_s$ and $d_t$ merely represent specific data dimensions and may not capture the heterogeneity in the nature or type of features between the source and target domains.
>
> **A.** Thanks for your suggestion, **we will revise Definition 1 by introducing the source and target feature spaces, i.e., $\mathcal{X}_s$ and $\mathcal{X}_t$. Under the SHDA setting, $\mathcal{X}_s \neq \mathcal{X}_t$, while in the SSDA problem, $\mathcal{X}_s = \mathcal{X}_t$**.
>
> >Q1. What was the rationale behind using fixed parameter settings for different SHDA tasks on the same dataset? Doesn't this approach risk not capturing the optimal performance for each task?
>
> **A.** Indeed, as suggested in the original papers of those baselines, they utilize the same parameter setting across different SHDA tasks on the same dataset. Consequently, adopting such settings could ensure optimal performance to some extent. **In our experiments, we solely modify the factors related to source samples, e.g., the number of source samples, to construct distinct SHDA tasks**. Thus, **to ensure a fair comparison, we adhere to the control variable principle, and maintain a consistent parameter setting for each baseline across different tasks on the same dataset**.
>
> >Q2. In the section of analysis on the original discriminability of source samples, what led to the choice of $\lambda = 0.4$ and $\lambda = 0.41$?
>
> **A.** We are sorry for this confusion. In the experiments, we design two strategies, i.e., **Category Replicate** and **Category Shift**, to quantitatively explore how the original discriminability of source samples affects the performance of SHDA. In the former strategy, we generate four noise domains, i.e., **NK$_3$**, **NK$_4$**, **NK$_5$**, and **NK$_6$**. Their LDA values are 40.51, 51.55, 55.77, and 60.57, respectively. **We find that all SHDA methods tend to perform better as the LDA values increase, which seems reasonable**. To further illustrate the generality of this conclusion, we utilize the latter strategy to empirically create two noise domains, making their LDA values approach to those of **NK$_5$** and **NK$_6$**, respectively. Accordingly, we empirically set the values of $\lambda$ to 0.40 and 0.41, thereby creating the noise domains of **NL$_{0.40}$** and **NL$_{0.41}$** with LDA values of 57.64 and 60.56, respectively. **However, building upon their experimental results, we find that the above conclusion does not hold**. Therefore, we construct two counterexamples to empirically demonstrate that the original discriminability of source samples is not the primary source of transferable knowledge in SHDA tasks. We will add the above statement in the revision.
>
> >Q3. In the section of analysis on the transferable discriminability of source samples, when using $g_t (\cdot)$ as a single layer fully connected networks with the Leaky ReLU, is there any potential for underfitting the data?
>
> **A.** Since HCN is trained using extracted features (e.g., $DeCAF_6$) rather than raw samples, even with the adoption of lightweight networks, under-fitting is unlikely to occur.

---

> > ### Comment · Reviewer_WVvA · 2023-11-22
> >
> > Thank you for the additional clarification and for updating the manuscript. I consider this work to be a valuable contribution to the field of domain adaptation, and I am inclined to maintain my original score.

---

> ### Author Response · Authors · 2023-11-23
> **Response to Reviewer WVvA**
>
> >Thank you for the additional clarification and for updating the manuscript. I consider this work to be a valuable contribution to the field of domain adaptation, and I am inclined to maintain my original score.
>
> **A.** Thanks again for affirming the value of our work and we appreciate it!

---

### Official Review · Reviewer_SDg6 · 2023-11-02

**Soundness:** 2 fair
**Presentation:** 3 good
**Contribution:** 2 fair
**Rating:** 3
**Confidence:** 3

**Summary:**

This paper conducts substantial empirical experiments to explore the effects of the number of source samples, the dimensions of source samples, the original discriminability of source samples, and the transferable discriminability of source samples to semi-supervised heterogeneous domain adaptation. However, I don’t think the experiment results can fully support the conclusions.

**Strengths:**

1. The paper is well-written and easy to understand.
2. The work of this paper is substantial.

**Weaknesses:**

I do not think the experiment results can support the conclusions due to the following concerns:
1. I think the experiment of “label information” is meaningless. There is no doubt that the order of category indices would not affect the performance since they are just symbols without any semantics. Additionally, I do not think label information can be regarded as category indices.
2. In Figure 3, the performance of SSAN changes significantly when the feature dimension changes, making the conclusion that feature information is not the dominant factor not convincing.
3. In the experiment of Table 1, I think the method of not adopting transfer learning should be included since both true source samples and noises may be unhelpful in this experiment.
4. In section 6, only one target domain is tested.
5. In the experiment of “original discriminability”, I do not think category replicate and category shift are appropriate. Category replicate assigns the same category label to different category samples, which would damage the training. This damage is more serious when K is larger. So, we can not know whether the discriminability or damage causes the effect. Category shift does not alter the internal distribution. Instead, I regard the Gaussian with different means and variances to be better. Specifically, larger mean and variance differences between categories represent larger discriminability. Additionally, the conclusion that the primary source of transferable knowledge in SHDA tasks does not lie in the original discriminability of source samples is not convincing since the performance improves when LDA values increase in NK3,4,5 and 6.
6. Why the metrics for measuring discriminability are different in the experiments of “original discriminability” and “transferable discriminability”? Concretely, one is LDA values, and the other is empirical risk.
7. In Table 2, why report the average accuracy of seven methods instead of individual results as in previous experiments? It makes me feel suspicious.
8. The authors consider the noises to be transferable and the key factor to be the transferable discriminability. However, the comparison of the transferable discriminability of true samples and noises are not given in Table 2 and 3.

**Questions:**

Please see the Weaknesses part.

---

> ### Author Response · Authors · 2023-11-20
> **Response to Reviewer SDg6 (Part I)**
>
> Thanks very much for your valuable comments and questions. We address your concerns as follows.
>
> >W1. I think the experiment of “label information” is meaningless. There is no doubt that the order of category indices would not affect the performance since they are just symbols without any semantics. Additionally, I do not think label information can be regarded as category indices.
>
> **A.** We feel that there may be a misunderstanding about the study on label information of source samples. **We want to emphasize that we keep the order of category indices unchanged for target samples and only modify them for source samples** (please refer to Figure 1 in our manuscript). As an example, **let's assume that there are three categories, i.e., cat, dog, and horse, and their ground-truth category indices are 1, 2, and 3, respectively**. **In the target domain, we keep the ground-truth category indices unchanged**. However, in the source domain, we randomly change the category index of samples belonging to the same category to that of another category. For instance, **we change the category indices of cat, dog, and horse to 2, 3, and 1, respectively**. As a result, **there is no one-to-one correspondence between the categories of the source and target domains**. Intuitively, this can affect target performance, especially in homogeneous scenarios where the source and target samples share the same feature extractor. In the SHDA, However, based on our experimental results, we observe that as the orders of category indices for source samples change, the accuracies of all methods remain almost unchanged. **This is a surprising and crucial finding**. Building upon this, we perform subsequent cross-dataset experiments to explore the impact of feature information on SHDA performance. **In those experiments, the categories of the source and target samples are completely distinct**. In addition, we can modify label information to category information in the revision.
>
> >W2. In Figure 3, the performance of SSAN changes significantly when the feature dimension changes, making the conclusion that feature information is not the dominant factor not convincing.
>
> **A.** **The principles of different methods are distinct, and there may be variations in performance trends. Hence, we observe the overall performance trends of all methods as source domains change, rather than focusing on individual ones**. Although SSAN shows relatively large fluctuations in performance, most methods exhibit relatively stable performance. Thus, those results can imply that the feature information of source samples is not the dominant factor affecting the performance of SHDA. We will explain it more clearly in the revision.
>
> >W3. In the experiment of Table 1, I think the method of not adopting transfer learning should be included since both true source samples and noises may be unhelpful in this experiment.
>
> **A.** In our experiments, SVMt and NNt are two supervised learning methods without any transfer mechanisms. Specifically, **SVMt and NNt only utilize the labeled target samples to train a support vector machine and a neural network, respectively**. Accordingly, we have reported the performance of non-transfer learning methods. Also, based on the results listed in Table 1, we can clearly observe that **most SHDA methods are far better than SVMt and NNt**. Specifically, on the task of A ($S_{800}$) $\rightarrow$ C ($D_{4096}$), the classification accuracy of STN is 88.41\%, which substantially exceeds the best supervised method, i.e., NNt, by 7.82\%. Furthermore, **it is difficult to achieve ideal performance solely through supervised learning approaches, which can be confirmed in many SHDA studies** (Please see details in [1-4]). Therefore, we have good reason to believe that using either noises or true source samples will contribute to improving target performance.
>
> **References**
>
> [1] Wei-Yu Chen, Tzu-Ming Harry Hsu, Yao-Hung Tsai, Yu-Chiang Frank Wang, and Ming-Syan Chen. Transfer neural trees for heterogeneous domain adaptation. In ECCV, 2016.
>
> [2] Yuan Yao, Yu Zhang, Xutao Li, and Yunming Ye. Heterogeneous domain adaptation via soft transfer network. In ACM MM, pp. 1578–1586, 2019.
>
> [3] Shuang Li, Binhui Xie, Jiashu Wu, Ying Zhao, Chi Harold Liu, and Zhengming Ding. Simultane- ous semantic alignment network for heterogeneous domain adaptation. In ACM MM, pp. 3866–3874, 2020.
>
> [4] Zhen Fang, Jie Lu, Feng Liu, and Guangquan Zhang. Semi-supervised heterogeneous domain adaptation: Theory and algorithms. TPAMI, 45(1):1087–1105, 2023.

---

> > ### Author Response · Authors · 2023-11-20
> > **Response to Reviewer SDg6 (Part II)**
> >
> > >W4. In section 6, only one target domain is tested.
> >
> > **A.** Thanks for your suggestion. Indeed, **based on Table 1 and Figure 3 in our manuscript, we have noticed that in addition to the target domain of **S**, a similar phenomenon exists in other target domains**. That is, using noises or features from other datasets as source samples can enhance the corresponding target performance. Accordingly, we can infer that our conclusions are valid on distinct target domains. In addition, to empirically confirm this, we also add a target domain, i.e., **C** ($D\_{4096}$), to quantitatively analyze the potential sources of transferable knowledge in the SHDA.
> >
> > Firstly, we generate 10 noise domains, i.e., **NS$\_{300}\^{10}$**, **NS$\_{400}\^{10}$**, **NS$\_{500}\^{10}$**, **NS$\_{600}\^{10}$**, **NS$\_{700}\^{10}$**, **ND$\_{100}\^{10}$**, **ND$\_{200}\^{10}$**, **ND$\_{300}\^{10}$**, **ND$\_{400}\^{10}$**, and **ND$\_{500}\^{10}$**. Here, **NS$\_{300}\^{10}$** and **NS$\_{300}$** are generated using the identical technique, yet **NS$\_{300}\^{10}$** consists of 10 categories while **NS$\_{300}$** contains six. Others are similar to this. In the following table, we find that the performance of all baselines is almost constant as the number and dimensions of source samples change.
> >
> > | $\mathcal{D}\_s \\rightarrow \mathcal{D}\_t$ | $\mathcal{TR}\_s$  | SVMt  | NNt | STN | SSAN  | JMEA |
> > | ---- | ---- | ---- | ---- | ---- | ---- | ---- |
> > | ND$\_{100}\^{10}$ $\\rightarrow$ C ($D\_{4096}$) | 0.57 | 79.61 | 80.59 | 86.59 | 86.22 | 86.12 |
> > | ND$\_{200}\^{10}$ $\\rightarrow$ C ($D\_{4096}$) | 0.51 | 79.61 | 80.59 | 87.15 | 87.78 | 85.55 |
> > | ND$\_{300}\^{10}$ $\\rightarrow$ C ($D\_{4096}$) | 0.48 | 79.61 | 80.59 | 85.24 | 87.42 | 85.73 |
> > | ND$\_{400}\^{10}$ $\\rightarrow$ C ($D\_{4096}$) | 0.48 | 79.61 | 80.59 | 87.16 | 87.18 | 85.54 |
> > | ND$\_{500}\^{10}$ $\\rightarrow$ C ($D\_{4096}$) | 0.46 | 79.61 | 80.59 | 86.81 | 87.09 | 85.54 |
> > | - | - | - | - | - | - | - |
> > | NS$\_{300}\^{10}$ $\\rightarrow$ C ($D\_{4096}$) | 0.48 | 79.61 | 80.59 | 85.86 | 86.94 | 85.88 |
> > | NS$\_{400}\^{10}$ $\\rightarrow$ C ($D\_{4096}$) | 0.49 | 79.61 | 80.59 | 85.35 | 86.94 | 85.96 |
> > | NS$\_{500}\^{10}$ $\\rightarrow$ C ($D\_{4096}$) | 0.48 | 79.61 | 80.59 | 85.24 | 87.42 | 85.73 |
> > | NS$\_{600}\^{10}$ $\\rightarrow$ C ($D\_{4096}$) | 0.48 | 79.61 | 80.59 | 85.26 | 86.84 | 86.13 |
> > | NS$\_{700}\^{10}$ $\\rightarrow$ C ($D\_{4096}$) | 0.49 | 79.61 | 80.59 | 85.31 | 87.22 | 86.04 |
> >
> > Secondly, we create eight noise domains, i.e., **NK$\_7\^{10}$**, **NK$\_8\^{10}$**, **NK$\_9\^{10}$**, **NK$\_{10}\^{10}$**, **NL$\_{0.25}\^{10}$**, **NL$\_{0.35}\^{10}$**, **NL$\_{0.45}\^{10}$** and **NL$\_{0.55}\^{10}$**. Here, the former four are generated using the **Category Replicate** strategy, whereas the latter four are constructed through the adoption of the **Category Shift** mechanism. Also, the superscript denotes the total number of categories, and the subscript denotes the corresponding parameter. In the following table, we can observe again that the original discriminability of source samples is not the dominant factor affecting the performance of SHDA. Also, **in conjunction with the above table, it is evident that transferable discriminability stands as the primary factor influencing the performance of SHDA**.
> >
> > | $\mathcal{D}\_s \\rightarrow \mathcal{D}\_t$ | LDA | $\mathcal{TR}\_s$ | SVMt | NNt | STN | SSAN | JMEA |
> > | ---- | ---- | ---- | ---- | ---- | ---- | ---- | ---- |
> > | NK$\_7\^{10}$ $\\rightarrow$ C ($D\_{4096}$) | 29.34 | 1.34 | 79.61 | 80.59 | 60.03 | 81.15 | 74.85 |
> > | NK$\_8\^{10}$ $\\rightarrow$ C ($D\_{4096}$) | 31.52 | 1.02 | 79.61 | 80.59 | 69.88 | 83.17 | 78.76 |
> > | NK$\_9\^{10}$ $\\rightarrow$ C ($D\_{4096}$) | 32.35 | 0.71 | 79.61 | 80.59 | 78.16 | 84.70 | 83.85 |
> > | NK$\_{10}\^{10}$ $\\rightarrow$ C ($D\_{4096}$) | **33.6** | **0.48** | **79.61** | **80.59** | **85.24** | **87.42** | **85.73** |
> > | - | - | - | - | - | - | - | - |
> > | NL$\_{0.25}\^{10}$ $\\rightarrow$ C ($D\_{4096}$) | 35.38 | 2.59 | 79.61 | 80.59 | 57.00 | 76.63 | 80.59 |
> > | NL$\_{0.35}\^{10}$ $\\rightarrow$ C ($D\_{4096}$) | 69.35 | 2.59 | 79.61 | 80.59 | 59.02 | 76.26 | 82.80 |
> > | NL$\_{0.45}\^{10}$ $\\rightarrow$ C ($D\_{4096}$) | 114.64 | 2.60 | 79.61 | 80.59 | 61.48 | 76.91 | 82.22 |
> > | NL$\_{0.55}\^{10}$ $\\rightarrow$ C ($D\_{4096}$) | 171.26 | 2.63 | 79.61 | 80.59 | 62.32 | 76.11 | 82.36 |

---

> > > ### Author Response · Authors · 2023-11-20
> > > **Response to Reviewer SDg6 (Part III)**
> > >
> > > >W5. In the experiment of “original discriminability”, I do not think category replicate and category shift are appropriate. Category replicate assigns the same category label to different category samples, which would damage the training. This damage is more serious when K is larger. So, we can not know whether the discriminability or damage causes the effect. Category shift does not alter the internal distribution. Instead, I regard the Gaussian with different means and variances to be better. Specifically, larger mean and variance differences between categories represent larger discriminability. Additionally, the conclusion that the primary source of transferable knowledge in SHDA tasks does not lie in the original discriminability of source samples is not convincing since the performance improves when LDA values increase in NK3,4,5 and 6.
> > >
> > > **A.** Firstly, there may be a misunderstanding about the **Category Replicate** strategy. **It first replicates $C - K$ copies of all samples from a specific category and then assigns the remaining categories to them individually**. Here, $C$ is the total number of categories in the target domain, and $K$ is the initial total number of categories in a noise domain. Thus, it does not assign the same category to different category samples.
> > >
> > > Secondly, based on our experiments, **we find that achieving higher target performance becomes remarkably easy by using different means and variances to generate noises**. Hence, it does not reflect the trend in target performance changing with the original discriminability of source samples very well. As a result, we design the **Category Replicate** strategy.
> > >
> > > Thirdly, we adopt the **Category Shift** strategy to design two counterexamples, i.e., **NL$\_{0.40}$** and **NL$\_{0.41}$**, which empirically demonstrate that the original discriminability of source samples is not the primary source of transferable knowledge in SHDA tasks. **Typically, if we encounter several counterexamples, then we can derive the above conclusion**.
> > >
> > > Fourthly, we can also construct other counterexamples. For instance, we create two noise domains, i.e., **NL$\_{0.50}$** and **NL$\_{1.00}$**, each with a respective value of $\lambda$ as 0.50 and 1.00. Also, their LDA values are 90.07 and 360.28, respectively. In the following table, we can clearly see that although the LDA values of **NL$\_{0.50}$** and **NL$\_{1.00}$** are much higher than that of **NK$\_6$**, the performance of SHDA baselines on the **NL$\_{0.50}$ $\\rightarrow$ S** and **NL$\_{1.00}$ $\\rightarrow$ S** tasks is lower than that on the **NK$\_6$ $\\rightarrow$ S** task. Also, the $\mathcal{TR}\_s$ values of the **NL$\_{0.50}$ $\\rightarrow$ S** and **NL$\_{1.00}$ $\\rightarrow$ S** tasks are higher than that of the **NK$\_{6}$ $\\rightarrow$ S** task. This once again implies that the primary source of transferable knowledge in SHDA tasks does not lie in the original discriminability of source samples.
> > >
> > > | $\mathcal{D}_s \rightarrow \mathcal{D}_t$ | LDA | $\mathcal{TR}\_s$ | SVMt | NNt | STN | SSAN | JMEA |
> > > | ---- | ---- | ---- | ---- | ---- | ---- | ---- | ---- |
> > > | NK$_5$ $\rightarrow$ S | 55.77 | 0.69 | 57.24 | 60.34 | 66.75 | 64.65 | 63.07 |
> > > | NK$_6$ $\rightarrow$ S | **60.57** | **0.27** | **57.24** | **60.34** | **69.91** | **67.11** | **69.33** |
> > > | - | - | - | - | - | - | - | - |
> > > | NL$_{0.40}$ $\rightarrow$ S | 57.64 | 1.73 | 57.24 | 60.34 | 63.82 | 60.24 | 64.92 |
> > > | NL$_{0.41}$ $\rightarrow$ S | 60.56 | 1.72 | 57.24 | 60.34 | 63.71 | 61.13 | 64.53 |
> > > | NL$_{0.50}$ $\rightarrow$ S | 90.07 | 1.70 | 57.24 | 60.34 | 64.94 | 59.41 | 64.29 |
> > > | NL$_{1.00}$ $\rightarrow$ S | 360.28 | 1.66 | 57.24 | 60.34 | 64.82 | 59.03 | 65.22 |
> > >
> > > **In summary, we have good reason to believe that our conclusion is convincing**.
> > >
> > > >W6. Why the metrics for measuring discriminability are different in the experiments of “original discriminability” and “transferable discriminability”? Concretely, one is LDA values, and the other is empirical risk.
> > >
> > > **A.** We are sorry for this confusion. Indeed, all of them can be used to measure the discriminability of source samples. **However, in the experiments, we find that the LDA values of source samples in the common space are very large. For instance, the LDA value of NK$\_6$ is 3912.9**. If we adopt the LDA values, we need to introduce a trade-off parameter for balancing the importance between the values of LDA and MMD. In addition, in the source space, there is no requirement for extra classifier training to calculate the empirical risk. Also, in the common space, there is no need for additional calculation of the LDA value. Both comply with the Occam's razor principle. Thus, we adopt the empirical risk of source samples to characterize their discriminability in the common space. We will make it more clearly in the revision.

---

> > > > ### Author Response · Authors · 2023-11-20
> > > > **Response to Reviewer SDg6 (Part IV)**
> > > >
> > > > >W7. In Table 2, why report the average accuracy of seven methods instead of individual results as in previous experiments? It makes me feel suspicious.
> > > >
> > > > **A.** **As aforementioned, the principles of different methods are distinct, and there may be variations in performance trends. Hence, we observe the overall performance trends of all methods as source domains change, rather than focusing on individual methods**. To this end, we report the average accuracy of seven SHDA methods. Thus, we do not repeatedly list their classification accuracies. In addition, in the revision, we will include those results in the appendix for supplementary clarification.
> > > >
> > > > | $\mathcal{D}_s \rightarrow \mathcal{D}_t$ | NS$\_{300}$ $\\rightarrow$ S | NS$\_{400}$ $\rightarrow$ S | NS$\_{500}$ $\rightarrow$ S | NS$\_{600}$ $\rightarrow$ S | NS$\_{700}$ $\rightarrow$ S | ND$\_{100}$ $\rightarrow$ S | ND$\_{200}$ $\rightarrow$ S | ND$\_{300}$ $\rightarrow$ S | ND$\_{400}$ $\rightarrow$ S | ND$\_{500}$ $\rightarrow$ S |
> > > > | ---- | ---- | ---- | ---- | ---- | ---- | ---- | ---- | ---- | ---- | ---- |
> > > > | $\mathcal{TR}\_s$ | 0.27 | 0.27 | 0.27 | 0.28 | 0.27 | 0.31 | 0.28 | 0.27 | 0.27 | 0.26 |
> > > > | HCN$\_{acc}$ | 56.10 | 55.79 | 56.07 | 55.87 | 56.00 | 55.88 | 56.30 | 56.07 | 56.24 | 55.94 |
> > > > | SVMt | 57.24 | 57.24 | 57.24 | 57.24 | 57.24 | 57.24 | 57.24 | 57.24 | 57.24 | 57.24 |
> > > > | NNt | 60.34 | 60.34 | 60.34 | 60.34 | 60.34 | 60.34 | 60.34 | 60.34 | 60.34 | 60.34 |
> > > > | SHFA | 63.69 | 63.65 | 63.65 | 63.66 | 63.65 | 63.66 | 63.62 | 63.65 | 63.62 | 63.66 |
> > > > | CDLS | 63.52 | 63.57 | 63.56 | 63.57 | 63.60 | 63.54 | 63.53 | 63.56 | 63.64 | 63.58 |
> > > > | DDACL | 63.63 | 63.35 | 63.92 | 63.63 | 63.22 | 63.78 | 63.42 | 63.92 | 64.06 | 63.89 |
> > > > | STN | 69.69 | 70.53 | 69.91 | 69.94 | 69.88 | 70.98 | 69.11 | 69.91 | 69.65 | 68.57 |
> > > > | SSAN | 67.49 | 67.07 | 67.11 | 67.13 | 67.15 | 66.99 | 67.37 | 67.11 | 68.80 | 67.63 |
> > > > | JMEA | 68.87 | 69.13 | 69.33 | 68.96 | 69.47 | 68.88 | 68.03 | 69.33 | 68.98 | 65.89 |
> > > >
> > > > | $\mathcal{D}_s \rightarrow \mathcal{D}_t$ | NK$\_3$ $\rightarrow$ S | NK$\_4$ $\rightarrow$ S | NK$\_5$ $\rightarrow$ S | NK$\_6$ $\rightarrow$ S | NL$\_{0.40}$ $\rightarrow$ S | NL$\_{0.41}$ $\rightarrow$ S |
> > > > | ---- | ---- | ---- | ---- | ---- | ---- | ---- |
> > > > | $\mathcal{TR}\_s$ | 1.87 | 1.22 | 0.69 | 0.27 | 1.73 | 1.72 |
> > > > | HCN$\_{acc}$ | 56.25 | 56.31 | 55.81 | 56.07 | 55.74 | 55.56 |
> > > > | SVMt | 57.24 | 57.24 | 57.24 | 57.24 | 57.24 | 57.24 |
> > > > | NNt | 60.34 | 60.34 | 60.34 | 60.34 | 60.34 | 60.34 |
> > > > | SHFA | 48.32 | 49.50 | 56.13 | 63.65 | 53.65 | 55.80 |
> > > > | CDLS | 61.82 | 62.95 | 63.89 | 63.56 | 59.19 | 59.20 |
> > > > | DDACL | 51.42 | 53.73 | 57.20 | 63.92 | 38.78 | 39.04 |
> > > > | STN | 58.41 | 60.95 | 66.75 | 69.91 | 63.82 | 63.71 |
> > > > | SSAN | 59.09 | 61.28 | 64.65 | 67.11 | 60.24 | 61.13 |
> > > > | JMEA | 59.27 | 62.57 | 63.07 | 69.33 | 64.92 | 64.53 |
> > > >
> > > > | $\mathcal{D}_s \rightarrow \mathcal{D}_t$ | NG $\rightarrow$ S | NU $\rightarrow$ S | NL $\rightarrow$ S |
> > > > | ---- | ---- | ---- | ---- |
> > > > | $\mathcal{TR}\_s$ | 0.26 | 0.55 | 0.54 |
> > > > | HCN$\_{acc}$ | 55.88 | 55.73 | 55.81 |
> > > > | SVMt | 57.24 | 57.24 | 57.24 |
> > > > | NNt | 60.34 | 60.34 | 60.34 |
> > > > | SHFA | 63.69 | 64.14 | 64.18 |
> > > > | CDLS | 63.46 | 58.69 | 58.63 |
> > > > | DDACL | 64.77 | 66.07 | 65.71 |
> > > > | STN | 69.29 | 68.71 | 69.37 |
> > > > | SSAN | 65.10 | 64.30 | 63.55 |
> > > > | JMEA | 66.79 | 68.66 | 69.15 |
> > > >
> > > > >W8. The authors consider the noises to be transferable and the key factor to be the transferable discriminability. However, the comparison of the transferable discriminability of true samples and noises are not given in Table 2 and 3.
> > > >
> > > > **A.** Thanks for your suggestion. In the following table, we provide a comparison of the transferable discriminability between true samples and noises. From the results, **we can clearly see that the $\mathcal{TR}\_s$ values of all noises except NK$\_3$, NK$\_4$, NK$\_5$, NL$\_{0.40}$, and NL$\_{0.41}$ are close to those of true samples**. This once again confirms the significance of the transferable discriminability. We will include those results in the appendix.
> > > >
> > > > | $\mathcal{D}_s \rightarrow \mathcal{D}_t$ | NS$\_{300}$ $\rightarrow$ S | NS$\_{400}$ $\rightarrow$ S | NS$\_{500}$ $\rightarrow$ S | NS$\_{600}$ $\rightarrow$ S | NS$\_{700}$ $\rightarrow$ S | ND$\_{100}$ $\rightarrow$ S | ND$\_{200}$ $\rightarrow$ S | ND$\_{300}$ $\rightarrow$ S | ND$\_{400}$ $\rightarrow$ S | ND$\_{500}$ $\rightarrow$ S |
> > > > | ---- | ---- | ---- | ---- | ---- | ---- | ---- | ---- | ---- | ---- | ---- |
> > > > | $\mathcal{TR}\_s$ | 0.27 | 0.27 | 0.27 | 0.28 | 0.27 | 0.31 | 0.28 | 0.27 | 0.27 | 0.26 |
> > > > | $\mathcal{D}_s \rightarrow \mathcal{D}_t$ | **NK3 $\rightarrow$ S** | **NK4 $\rightarrow$ S** | **NK5 $\rightarrow$ S** | **NK6 $\rightarrow$ S** | **NL$\_{0.40}$ $\rightarrow$ S** | **NL$\_{0.41}$ $\rightarrow$ S** | **E $\rightarrow$ S** | **F $\rightarrow$ S** | **G $\rightarrow$ S** | **I $\rightarrow$ S** |
> > > > | $\mathcal{TR}\_s$ | 1.87 | 1.22 | 0.69 | 0.27 | 1.73 | 1.72 | 0.25 | 0.26 | 0.27 | 0.26 |

---

> ### Comment · Reviewer_SDg6 · 2023-11-21
>
> Thank you for your responses. However, your replys do not solve my concerns well. So, I would like to maintain my original scores.
> 1. In the example you mentioned, the category oder affects the performance since both the feature extractor and classification head are transferred. However, since only the feature extractor is reused in your experiments while the classification head is discarded , the category order does not affect. This has nothing to do with homogeneity or heterogeneity. I insist that the experiment of category order is not meaningful.
> 2. I can not agree with your opinion. In addition to SSAN, the changes of JMEA and DDACL are not negligible.
> 3. SVMt and NNt are two weak baselines. I would like to know the performances of the SHDA methods in Table 1 without using true source samples or noises.
> 4. Assigning a sample with two (or more) different category labels would also damage the training. We can not know whether the discriminability or damage causes the effect.
> 5. Additionally, the conclusion that the primary source of transferable knowledge in SHDA tasks does not lie in the original discriminability of source samples is not convincing since the performance improves when LDA values increase in NK3,4,5 and 6.
> 6. ''We find that achieving higher target performance becomes remarkably easy by using different means and variances to generate noises.'' This phenomenon doest no support your conclusion about original discriminability.
> 7. Moreover, category replicate and category shift can not fully represent original discriminability.
> 8. I suggest to apply the same metric for measuring the two discriminability.

---

> ### Author Response · Authors · 2023-11-23
> **Response to Reviewer SDg6 (Part I)**
>
> Thanks for your patience and meticulous response. We address your concerns as follows.
>
> >W1. In the example you mentioned, the category order affects the performance since both the feature extractor and classification head are transferred. However, since only the feature extractor is reused in your experiments while the classification head is discarded , the category order does not affect. This has nothing to do with homogeneity or heterogeneity. I insist that the experiment of category order is not meaningful.
>
> **A1.** We feel that there may be a misunderstanding about the training manner of **Semi-supervised Heterogeneous** Domain Adaptation (SHDA) (please refer to Section 3 in the manuscript for the problem definition of SHDA). Unlike **source-free** domain adaptation, **SHDA does not begin by pre-training a feature extractor and then fine-tuning it by using both labeled and unlabeled target samples**. In SHDA, all approaches typically follow a two-step process. They first pre-process source and target samples by extracting heterogeneous features, e.g., 800-dimension $SURF$ ($S_{800}$) and 4096-dimension $DeCAF_6$ ($D_{4096}$), and then perform **heterogeneous** domain adaptation by **jointly training the source and target features from scratch. Assuming that the source and target features are homogeneous and share the same feature projection, we believe that the order of category indices of source features will affect target performance**.
>
> To empirically verify this, we have conducted an additional experiment on the **S** domain. Specifically, we randomly and evenly partition all samples into the source and target domains, i.e., **S$\_s$** and **S$\_t$**. For the source domain, we utilize all samples as labeled ones. As for the target domain, we randomly sample 3 samples per category as the labeled ones, and the rest of the samples are considered as the unlabeled ones. **This is a homogeneous scenario where the source and target samples are from the same feature space. Following the setting in our manuscript, for source samples within the same category, we randomly change their category index to correspond to a distinct category**. The table below lists the performance of NN, which learns a neural network by using all labeled source and target samples. **Note that in NN, a shared feature projection for both source and target samples is learned from scratch**. We can see that when the category indices of source samples do not follow the ground-truth order, i.e., order 1, the target performance is extremely poor. **This implies that the order of category indices for source samples is important in such scenarios**. This is in line with our expectation.
>
> | S$\_s$ $\rightarrow$ S$\_t$ |  NN |
> | ---- | ---- |
> | Order1 | 83.92 |
> | Order2 | 10.92 |
> | Order3 | 7.80  |
> | Order4 | 8.32  |
> | Order5 | 6.84  |
> | Order6 | 10.98 |
>
> **In our experiments, however, we find that the target performance is not affected by the order of category indices of source features**. The primary reason is that **in SHDA, due to the heterogeneity of source and target features, most approaches utilize different feature projections for them**. Accordingly, **we believe that this finding exhibits distinctive characteristics of SHDA, rather than being a meaningless finding**. We will make it more clearly in the revision.
>
> >W3. SVMt and NNt are two weak baselines. I would like to know the performances of the SHDA methods in Table 1 without using true source samples or noises.
>
> **A3.** Similar to W1, we believe that there may be a misunderstanding regarding the training manner of SHDA. Please refer to A1 for details. **To the best of our knowledge, all SHDA approaches require source features for classifier training and distribution alignment**. As an example, **in SHDA, a simple and common distribution alignment mechanism is to align the global and category centroids of the source and target features during adaptation**. Therefore, they cannot be directly extended to scenarios without source features.

---

> ### Author Response · Authors · 2023-11-23
> **Response to Reviewer SDg6 (Part II)**
>
> >W2. I can not agree with your opinion. In addition to SSAN, the changes of JMEA and DDACL are also negligible.
>
> **A2.** To further explain this, we can calculate the **coefficient of variation** (CV). **It is a statistical measure used to assess the relative variability, defined as the ratio of the standard deviation (STD) $\sigma$ to the mean $\mu$** [1]. The following tables list the raw data used to draw Figure 3 in our manuscript. Based on those results, **we can clearly see that the maximum value of CV for SSAN is 0.04, which is generally considered a relatively small value mathematically**. In the revision, we will incorporate those results and discussions into the Appendix for supplementary clarification.
>
> | $\mathcal{D}\_s \rightarrow \mathcal{D}\_t$ | SVMt  | NNt   | SHFA  | CDLS  | DDACL | STN   | SSAN  | JMEA |
> | ---- | ---- | ---- | ---- | ---- | ---- | ---- | ---- | ---- |
> | E  $\rightarrow$  S  | 57.24 | 60.34 | 63.53 | 58.98 | 65.82 | 70.05 | 66.30 | 68.50 |
> | F  $\rightarrow$  S  | 57.24 | 60.34 | 63.54 | 59.25 | 66.11 | 70.12  | 67.20 | 69.34 |
> | G $\rightarrow$ S   | 57.24 | 60.34 | 63.40 | 59.32 | 65.83 | 70.00 | 67.90 | 68.62 |
> | I  $\rightarrow$  S  | 57.24 | 60.34 | 63.58 | 59.07 | 66.17 | 70.28 | 66.70 | 69.10 |
> | A ($S\_{800}$)  $\rightarrow$  S | 57.24 | 60.34 | 64.04 | 60.40 | 64.17  | 68.03 | 69.28  | 67.76  |
> | C ($S\_{800}$)  $\rightarrow$  S | 57.24 | 60.34 | 64.11 | 59.75 | 63.70  | 67.58 | 63.95  | 67.94  |
> | W ($S\_{800}$)  $\rightarrow$  S | 57.24 | 60.34 | 63.95 | 61.83 | 65.79  | 68.25 | 67.20  | 68.00  |
> | A ($D\_{4096}$)  $\rightarrow$  S | 57.24 | 60.34 | 63.99 | 62.53 | 65.44  | 68.81 | 66.46  | 68.08  |
> | C ($D\_{4096}$)  $\rightarrow$  S | 57.24 | 60.34 | 63.94 | 60.87 | 64.65  | 67.58 | 66.19  | 68.33  |
> | W ($D\_{4096}$)  $\rightarrow$  S | 57.24 | 60.34 | 63.75 | 62.38 | 65.92  | 67.86 | 67.31  | 68.38  |
> | Text  $\rightarrow$  S | 57.24 | 60.34 | 63.81 | 62.59 | 64.58  | 67.97 | 66.12  | 68.44 |
> | - | - | - | - | - | - | - | - | - |
> | MEAN   | 57.24 | 60.34 | 63.79 | 60.63 | 65.29 | 68.78 | 66.78 | 68.41 |
> | STD   | 0.00  | 0.00  | 0.24  | 1.47  | 0.86  | 1.11  | 1.31  | 0.48 |
> | CV    | 0.00  | 0.00  | 0.00  | 0.02  | 0.01  | 0.02  | 0.02  | 0.01 |
>
> | $\mathcal{D}\_s \rightarrow \mathcal{D}\_t$ | SVMt | NNt | SHFA | CDLS | DDACL | TNT | STN | SSAN | JMEA |
> | ---- | ---- | ---- | ---- | ---- | ---- | ---- | ---- | ---- | ---- |
> | T  $\rightarrow$  Image | 67.93 | 68.77 | 69.40 | 71.68 | 75.69 | 78.18 | 78.21 | 76.89 | 78.91 |
> | A (S800)  $\rightarrow$  Image | 67.93 | 68.77 | 69.80 | 69.95 | 73.99 | 73.75 | 77.19 | 76.33 | 77.99 |
> | C (S800)  $\rightarrow$  Image | 67.93 | 68.77 | 69.73 | 69.44 | 73.05 | 75.36 | 76.25 | 75.56 | 77.58 |
> | W (S800)  $\rightarrow$  Image | 67.93 | 68.77 | 69.80 | 70.93 | 72.25 | 74.19 | 75.86 | 77.00 | 79.57 |
> | A (D4096)  $\rightarrow$  Image | 67.93 | 68.77 | 69.80 | 71.90 | 74.69 | 76.65 | 78.50 | 73.14 | 76.90 |
> | C (D4096)  $\rightarrow$  Image | 67.93 | 68.77 | 69.71 | 70.69 | 74.73 | 77.05 | 76.89 | 69.93 | 80.06 |
> | W (D4096)  $\rightarrow$  Image | 67.93 | 68.77 | 69.65 | 71.63 | 73.19 | 76.38 | 75.71 | 73.10 | 79.34 |
> | - | - | - | - | - | - | - | - | - |
> | MEAN   | 67.93 | 68.77 | 69.70 | 70.89 | 73.94 | 75.94 | 76.94 | 74.56 | 78.62 |
> | STD   | 0.00  | 0.00  | 0.14  | 0.93  | 1.19  | 1.59  | 1.10  | 2.62  | 1.16 |
> | CV    | 0.00  | 0.00  | 0.00  | 0.01  | 0.02  | 0.02  | 0.01  | 0.04  | 0.01 |
>
> **References**
>
> [1] https://en.wikipedia.org/wiki/Coefficient_of_variation

---

> ### Author Response · Authors · 2023-11-23
> **Response to Reviewer SDg6 (Part III)**
>
> >W4. Assigning a sample with two (or more) different category labels would also damage the training. We can not know whether the discriminability or damage causes the effect. Additionally, the conclusion that the primary source of transferable knowledge in SHDA tasks does not lie in the original discriminability of source samples is not convincing since the performance improves when LDA values increase in NK3,4,5 and 6. ''We find that achieving higher target performance becomes remarkably easy by using different means and variances to generate noises.'' This phenomenon does not support your conclusion about original discriminability. Moreover, category replicate and category shift can not fully represent original discriminability. I suggest to apply the same metric for measuring the two discriminability.
>
> **A4.** Thanks for your suggestion. **To address your concerns and further confirm our conclusion**, we have conducted additional experiments to examine how the original discriminability of source samples affects the performance of SHDA. Specifically, we first randomly generate $C$ different means and variances, i.e., $\boldsymbol{\mu\_1}$, $\boldsymbol{\mu\_2}$, $\cdots$, $\boldsymbol{\mu}\_C$, and $\mathbf{\Sigma}\_1$, $\mathbf{\Sigma}\_2$, $\cdots$, $\mathbf{\Sigma}\_C$. Then, we multiply them by a coefficient of $\alpha$ to quantitatively control the magnitude of the discriminability, i.e., $\boldsymbol{\mu}\_1 \times \mathbf{1} \times 1 \alpha$, $\boldsymbol{\mu}\_2 \times \mathbf{1} \times 2 \alpha$, $\cdots$, $\boldsymbol{\mu}\_C \times \mathbf{1} \times C \alpha$, and $\mathbf{\Sigma}\_1 \times \mathbf{1} \times 1 \alpha$, $\mathbf{\Sigma}\_2 \times \mathbf{1} \times 2 \alpha$, $\cdots$, $\mathbf{\Sigma}\_C \times \mathbf{1} \times C \alpha$. Here, $\mathbf{1}$ denotes an all-one matrix with an appropriate size. Accordingly, we create different noise domains by setting distinct values of $\alpha$. Generally, the larger the value of $\alpha$, the greater the discriminability. As a result, we construct 10 noise domains, i.e., **NA**$\_{0.2}^6$, **NA**$\_{0.4}\^6$, **NA**$\_{0.6}\^6$, **NA**$\_{0.8}\^6$, **NA**$\_{1.0}\^6$, **NA**$\_{0.2}\^{10}$, **NA**$\_{0.4}\^{10}$, **NA**$\_{0.6}\^{10}$, **NA**$\_{0.8}\^{10}$, and **NA**$\_{1.0}\^{10}$.
> Here, the superscript denotes the total number of categories, and the subscript denotes the value of $\alpha$. Moreover, **we adopt the empirical risk of source samples to measure the original discriminability**. To this end, we first randomly select 10\% of samples from each category as training samples, while the rest are assigned as testing samples. Then, we use those training samples to train a neural network and report the empirical risk on testing ones, i.e., $\mathcal{R}\_s\^o$.
>
> The table below reports the average results of 10 random trials. Note that due to the relatively small values of $\mathcal{R}\_s\^o$, we multiply them by 10 for comparison with those of $\mathcal{TR}\_s$. According to the results, **we find that the performance of all approaches is almost constant as the values of $\mathcal{R}\_s\^o$ decline**. This once again demonstrates that the original discriminability of source samples is not the primary factor influencing the performance of SHDA. Also, we can see that the change in the values of $\mathcal{TR}\_s$ is smaller than that of $\mathcal{R}\_s\^o$. According to those results, we have good reason to believe that the transferable discriminability of source samples stands as the dominant factor affecting the performance of SHDA.
>
> | $\mathcal{D}\_s \rightarrow \mathcal{D}\_t$ | $\mathcal{R}\_s\^{o}$ (x 10) | $\mathcal{TR}\_s$ | STN | SSAN | JMEA |
> | ---- | ---- | ---- | ---- | ---- | ---- |
> | NA$\_{0.2}\^6$ $\rightarrow$ S | 0.55  | 0.32  | 70.57  | 67.57  | 67.75  |
> | NA$\_{0.4}\^6$ $\rightarrow$ S | 0.17  | 0.29  | 70.51  | 68.80  | 68.05  |
> | NA$\_{0.6}\^6$ $\rightarrow$ S | 0.09  | 0.28  | 70.71  | 67.51  | 68.45  |
> | NA$\_{0.8}\^6$ $\rightarrow$ S | 0.07  | 0.27  | 69.98  | 67.05  | 68.18  |
> | NA$\_{1.0}\^6$ $\rightarrow$ S | 0.06  | 0.28  | 70.12  | 66.79  | 68.20  |
> | - | - | - | - | - | - |
> | NA$\_{0.2}\^{10}$ $\rightarrow$ TCD | 0.60  | 0.59  | 87.00  | 87.97  | 85.75  |
> | NA$\_{0.4}\^{10}$ $\rightarrow$ TCD | 0.25  | 0.53  | 86.65  | 87.17  | 85.89  |
> | NA$\_{0.6}\^{10}$ $\rightarrow$ TCD | 0.11  | 0.50  | 85.99  | 87.75  | 85.76  |
> | NA$\_{0.8}\^{10}$ $\rightarrow$ TCD | 0.10  | 0.49  | 85.54  | 87.73  | 85.30  |
> | NA$\_{1.0}\^{10}$ $\rightarrow$ TCD | 0.09  | 0.48  | 85.77  | 87.58  | 85.39  |
>
> We will utilize this experiment to replace the previous one. In addition, due to time constraints, in the above experiment, we mainly explore several advanced SHDA baselines, i.e., STN, SSAN, and JMEA, to once again verify that our conclusions are convincing. The results of other SHDA baselines will be included in the camera-ready version.

---

### Official Review · Reviewer_AqYr · 2023-11-05

**Soundness:** 2 fair
**Presentation:** 2 fair
**Contribution:** 2 fair
**Rating:** 3
**Confidence:** 5

**Summary:**

This study conducted a comprehensive empirical investigation of Semi-supervised Heterogeneous Domain Adaptation (SHDA) on seven SHDA approaches across massive SHDA tasks. Based on experiment results, authors find that the noises drawn from simple distributions are transferable across domains. Further investigation shows that transferable discriminability of source samples is vital for SHDA.

**Strengths:**

- It's the first to conduct an empirical study investigating the SHDA problem. Comprehensive and detailed experiments are conducted.

- The paper identifies and demonstrates that noises drawn from simple distributions can be effectively transferred to target. This finding opens up new possibilities for future work direction.

- Authors reveal the primary role of transferable discriminability of source samples.

**Weaknesses:**

- The study is biased so the conclusion drawn from the study might not be generalizable. The features are precomputed by some descriptors  and are not learnable. However, many semi-supervised domain adaptation methods, especially those based on deep learning, are powerful because they learn the feature extraction networks to generate discriminative features. If the feature is fixed, the real power of these method cannot be realized, and the studied based on this is not generalizable.

- The effect of the number of labeled target samples has not been studied. For semi-supervised domain adaptation, the number of labeled target samples is a crucial factor for the adaptation performance. However, this study does not cover the investigation on this aspect.

 - The value of this study is kind of limited. Semi-supervised heterogeneous domain adaptation is a very small field (as can be seen from the literature) and a study on such a small field can only draw attention on a small group of audience. While I agree this is valuable, the value is not very significant. We expect a study paper accepted at this conference to be of high scope so that a broad group of people can learn something from the study.

**Questions:**

See the weakness above.

---

> ### Author Response · Authors · 2023-11-20
> **Response to Reviewer AqYr**
>
> Thanks very much for your valuable comments and questions. We address your concerns as follows.
>
> >W1. The study is biased so the conclusion drawn from the study might not be generalizable. The features are precomputed by some descriptors and are not learnable. However, many semi-supervised domain adaptation methods, especially those based on deep learning, are powerful because they learn the feature extraction networks to generate discriminative features. If the feature is fixed, the real power of these methods cannot be realized, and the study based on this is not generalizable.
>
> **A.** We feel that there may be a misunderstanding about the research motivation. Please refer to the general response for details.
>
> >W2. The effect of the number of labeled target samples has not been studied. For semi-supervised domain adaptation, the number of labeled target samples is a crucial factor for the adaptation performance. However, this study does not cover the investigation on this aspect.
>
> **A.** **We want to emphasize that, in this study, we do not propose a new SHDA approach. Instead, our focus is on exploring the essence of the SHDA**, i.e., What knowledge from a heterogeneous source domain is transferred to the target domain? Accordingly, we adhere to the control variable principle. For the same target domain, we explore the above problem by keeping the number of target samples unchanged while changing the source domain. Also, based on our experience in the SHDA, we believe that, within a certain range, as the number of target samples increases, it will only enhance the performance of all baselines without affecting our current conclusions. **Hence, we do not conduct experiments involving variations in the number of labeled target samples**.
>
> In addition, to empirically verify this, we report the classification accuracies of some approaches w.r.t. distinct numbers of labeled target samples per category in the following table. We can clearly observe that their performance is boosted with the increase in the number of labeled target samples. This is in line with our expectation. In the revision, we will include those results in the appendix for supplementary clarification.
>
> | $\mathcal{D}_s \rightarrow \mathcal{D}_t$ | $l$ | NNt | STN | SSAN | JMEA |
> | ---- | ---- | ---- | ---- | ---- | ---- |
> | N$_6$ $\rightarrow$ S | 5 | 60.34 | 69.91 | 67.11 | 69.33 |
> | N$_6$ $\rightarrow$ S | 10 | 68.02 | 72.57 | 72.86 | 73.04 |
> | N$_6$ $\rightarrow$ S | 15 | 73.02 | 76.44 | 76.05 | 76.34 |
> | N$_6$ $\rightarrow$ S | 20 | 74.57 | 78.47 | 77.55 | 78.66 |
>
> >W3. The value of this study is kind of limited. Semi-supervised heterogeneous domain adaptation is a very small field (as can be seen from the literature) and a study on such a small field can only draw attention on a small group of audience. While I agree this is valuable, the value is not very significant. We expect a study paper accepted at this conference to be of high scope so that a broad group of people can learn something from the study.
>
> **A.** **Thank you for acknowledging the value of our research. However, we respectfully and politely disagree that the value of research should be determined by the size of the field**. Please refer to the general response for details.

---

### Author Response · Authors · 2023-11-20
**General Response**

We wish to express our deep gratitude to the reviewers for their comments. This study builds upon our extensive research experience in Semi-supervised **Heterogeneous** Domain Adaptation (SHDA). Also, **we rigorously adhere to the control variable principle**.  We believe that this study is unbiased and holds significant value. Thus, **with all due respect, we think the reviewers may have biases and misunderstandings about this study**.

Firstly, we feel that there may be a misunderstanding about the research motivation. **This study is inspired by Semi-supervised Heterogeneous Domain Adaptation (SHDA) rather than Semi-supervised Homogeneous Domain Adaptation (SHoDA)**. The problem setting of SHDA differs from that of SHoDA. In the SHDA problem, all methods typically follow a two-step process. They first pre-process source and target samples by extracting heterogeneous features, e.g., 800-dimension $SURF$ ($S_{800}$) and 4096-dimension $DeCAF_6$ ($D_{4096}$), and then perform heterogeneous domain adaptation. As far, numerous advanced approaches have been developed [1-2], resulting in improved transfer performance across heterogeneous domains. However, **we note that heterogeneous features may be very dissimilar. Also, those approaches do not utilize several very deep pre-trained networks**. So we want to ask ``why can knowledge be effectively transferred across very dissimilar heterogeneous features?'' In other words, what knowledge from a heterogeneous source domain is transferred to the target domain? **Inspired by this, we conduct this study to delve deeper into the essence of the SHDA. Accordingly, we conclude that noises are transferable. Otherwise, we may not be able to discover it**.

Secondly, we have reported the performance of non-transfer learning methods, i.e., SVMt and NNt. They are two supervised learning baselines. Based on the experimental results, we can clearly see that most SHDA methods are far better than SVMt and NNt. Thus, **we have good reason to believe that noises are transferable, and their utilization will contribute to improving target performance in the SHDA**. Although the conclusion is derived from the SHDA, we believe that this study will inspire numerous interesting studies. Also, as we detailed in the Discussion section, researchers in other fields, i.e., representation learning [3] and federated learning [4-5], have paid attention to this surprising perspective. Therefore, **we believe that the conclusion, i.e., noises are transferable, is illuminating and generalizable**.

Thirdly, to further confirm our conclusions, we have conducted some additional experiments: (i) We have performed experiments involving variants in the number of labeled target samples. (ii) In section 6, we have added a target domain, i.e., C ($D_{4096}$), to quantitatively analyze the potential sources of transferable knowledge in the SHDA. (iii) We have included two extra counterexamples, i.e., **NL$_{0.50}$** and **NL$_{1.00}$**, to further demonstrate the original discriminability of source samples is not the primary source of transferable knowledge in the SHDA. In addition, **due to time constraints, in the above experiments, we mainly explore several advanced SHDA baselines, i.e., STN, SSAN, and JMEA, to once again verify that our conclusions are convincing. The results of other SHDA baselines will be included in the camera-ready version**.

Last but not least, we agree that SHDA is a small field, however, **we respectfully and politely disagree that the value of research should be determined by the size of the field**. Our belief is that groundbreaking research often emerges from perseverance in less mainstream domains. Hence, we choose to explore and contribute to those less-explored areas, and try our best to do several impactful studies. Also, we submit this study to ICLR, since we believe that it is an inclusive and unbiased conference. **We hope that, with the influence of ICLR, researchers from diverse fields could recognize the value of this study and draw inspiration from it**.

Below please find our detailed point-by-point responses to reviewers' comments. Please let us know if you have further concerns.

**References**

[1] Day O, Khoshgoftaar T M. A survey on heterogeneous transfer learning[J]. Journal of Big Data, 2017.

[2] Runxue Bao, Yiming Sun, Yuhe Gao, Jindong Wang, Qiang Yang, Haifeng Chen, Zhi-Hong Mao, and Ye Ye. (2023). A Survey of Heterogeneous Transfer Learning.

[3] Manel Baradad Jurjo, Jonas Wulff, Tongzhou Wang, Phillip Isola, and Antonio Torralba. Learning to see by looking at noise. In NeurIPS, 2021.

[4] Mi Luo, Fei Chen, Dapeng Hu, Yifan Zhang, Jian Liang, and Jiashi Feng. No fear of heterogeneity: Classifier calibration for federated learning with non-iid data. In NeurIPS, 2021.

[5] Zhenheng Tang, Yonggang Zhang, Shaohuai Shi, Xin He, Bo Han, and Xiaowen Chu. Virtual homogeneity learning: Defending against data heterogeneity in federated learning. In ICML, 2022.

---

### Meta-Review · Area_Chair_FUr7 · 2023-12-06

**Metareview:**

This paper presents an experimental study on Heterogeneous Domain Adaptation which shows that noises drawn from simple distribution can be transferable allowing one to improve the performance on the target domain. They also studied other factors that impact the transferable discriminability of source examples.

On the positive side, the reviews have highlighted that the paper studies noise transferability in the context of semi-supervised heterogeneous domain adaptation which is interesting and new, an extensive experimental evaluation is presented, the analysis provides interesting features.
On the negative side, the impact of the experimental study is not clear or at least its interest appears limited and does not fully support the conclusions, in particular the way the authors consider the models, and in particular the way they deal with the labels (or not) are not satisfactory. Since the paper is mainly experimental, this is a strong limitation.

The authors did a strong effort to provide a long and dense rebuttal, with many additional experiments.
Reviewers have agreed that the authors did a strong effort with the rebuttal. Reviewer not completely satisfied with the answers which did not address all their concerns. Other reviewers did not support the paper, the scope appears limited and while interesting the material provided in the rebuttal is so dense that another round review would be necessary to valid the modifications.

As a consequence, I propose rejection.
I encourage nevertheless the authors to continue the improvement of this work for other venues.


.

**Justification For Why Not Higher Score:**

2 reviewers gave an evaluation as reject and one experimented reviewer maintains its evaluation.

**Justification For Why Not Lower Score:**

N/A

---

### Decision · Program_Chairs · 2024-01-16

Reject